# Increasing occurrence of cold and warm extremes during the recent global warming slowdown

Nathaniel C. Johnson [1,2,3,4], Shang-Ping Xie[3], Yu Kosaka [3,5] & Xichen Li [3,6]

The recent levelling of global mean temperatures after the late 1990s, the so-called global warming hiatus or slowdown, ignited a surge of scientific interest into natural global mean surface temperature variability, observed temperature biases, and climate communication, but many questions remain about how these findings relate to variations in more societally relevant temperature extremes. Here we show that both summertime warm and wintertime cold extreme occurrences increased over land during the so-called hiatus period, and that these increases occurred for distinct reasons. The increase in cold extremes is associated with an atmospheric circulation pattern resembling the warm Arctic-cold continents pattern, whereas the increase in warm extremes is tied to a pattern of sea surface temperatures resembling the Atlantic Multidecadal Oscillation. These findings indicate that large-scale factors responsible for the most societally relevant temperature variations over continents are distinct from those of global mean surface temperature.

[1] Atmospheric and Oceanic Sciences Program, Princeton University, Princeton 08540 NJ, USA. [2] National Oceanic and Atmospheric Administration/ Geophysical Fluid Dynamics Laboratory, Princeton University Forrestal Campus, 201 Forrestal Road, Princeton, NJ 08540-6649, USA. [3] Scripps Institution of Oceanography, University of California, San Diego, 9500 Gilman Drive #0206, La Jolla, CA 92093-0206, USA. [4] International Pacific Research Center, SOEST, University of Hawaii at Manoa, Honolulu 96822 Hawaii, USA. [5] Research Center for Advanced Science and Technology, University of Tokyo, 4-6-1 Komaba, Meguro-ku, Tokyo 153-8904, Japan. [6] Institute of Atmospheric Physics, Chinese Academy of Sciences Chao Yang District, P.O. Box 9804, Beijing 100029, China. Correspondence and requests for materials should be addressed to N.C.J. (email: nathaniel.johnson@noaa.gov)

Since the start of the 21st century, there has been considerable scientific and media focus on numerous costly episodes of extreme temperatures, including extreme summer heatwaves in Europe in 2003, Russia and Japan in 2010, and North America in 2011[1]. These extreme heatwaves occurred during a period when annual global mean surface temperature (GMST) remained nearly steady, a period referred to as a hiatus or global warming-slowdown. Although the "hiatus" terminology has been debated[2,3], and sources of negative biases in some global temperature records have been identified[4,5], the decadal-timescale apparent levelling of the global mean temperature remains a robust feature with known physical mechanisms. Most notably, hiatus periods like the most recent occurrence, have been connected with equatorial Pacific cooling and increases in deep-ocean heat uptake[6–9]. These mechanisms, however, fail to provide a sufficient explanation for the continued rise of extreme hot temperature occurrence over land, especially measures of the most extreme occurrence, over the past 15–20 years[10].

During this same period, frequent occurrences of severe winter cold, including the cold and snowy winters of 2009/10, 2010/11, and 2013/14 over portions of Eurasia and North America[11,12], have generated a contradictory perception of increasing cold extremes. This perception of contrasting variations in extreme summer warmth and winter cold suggests a recent increase in winter-to-summer temperature contrasts over NH continents. Such an increase in temperature variability would contrast the observed and projected decrease in subseasonal[13] and year-to-year[14] temperature variance under increasing greenhouse gases in conjunction with the reduced pole-to-equator

The apparent contrast in behavior between GMST and continental extreme temperature occurrence during the hiatus period suggests that the leading mechanisms responsible for their variations may be distinct. Such distinctions imply that some mechanisms may preferentially warm or cool the NH land relative to the oceans or Southern Hemisphere, or that different mechanisms modulate the seasonal evolution of global temperature. Previous studies have highlighted notable regional and seasonal variations in temperature trends during the hiatus period[7,9,15], but it remains uncertain if the dominant mechanisms for decadal-timescale annual GMST variability, namely those related to equatorial Pacific surface temperature variability, also hold for regional and seasonal extreme temperature occurrence. The NH continents represent a relatively small fraction of the global surface area but account for a large fraction of the societal impacts of extreme temperature occurrence, and so we must place particular focus on differentiating these sources of regionally and seasonally varying extreme temperature occurrence from those of annual GMST.

In this study, we examine the observed changes in extreme temperature occurrence over the NH continents during the recent global warming slowdown period. Consistent with the perception noted above, this analysis indeed reveals that both summertime warm and wintertime cold extreme temperature occurrences increased from 2002 to 2014. Additional analyses of observational data and climate model simulations indicate that the drivers of these hemispheric extreme temperature changes relate to naturally occurring, large-scale climate patterns in the atmosphere and oceans, and that these patterns are distinct from those that are believed to be a primary driver of the global warming slowdown.

## Results

**Modeling observed changes in extreme temperature occurrence.** We first examine the observed changes in continental extreme temperature occurrence over the past several decades. Figure 1a provides the NH land areal mean counts of wintertime cold and summertime warm extreme temperature occurrence, defined as the number of days per season with maximum temperature anomalies below the 10th percentile (TX10d) and above the 90th percentile (TX90d), respectively, per season (Methods) from 1979–2014. Consistent with ref. [10], the frequency of summertime warm extremes over NH land has exhibited a nearly monotonic increase without any evidence of a pause. The frequency of NH land cold extremes, in contrast, exhibited a rapid decline until nearly 2000 before levelling and subsequently increasing through 2014. The average number of both warm and cold extremes increased during the hiatus period, which we define as 2002–2014, at a rate of 2.1 and 1.5 days per decade, respectively, and this general behavior is not sensitive to the definition of hiatus period (Methods). Recent trends in the Southern Hemisphere continents are similar (Supplementary Fig. 1), but we restrict our focus to NH land areas where most of the reliable observations reside.

In contrast to the expected response to anthropogenic forcing[16], the trends in cold and warm extremes during the hiatus period display substantial spatial heterogeneity (Fig. 1b,c). Cold extremes generally have increased most over midlatitude continental regions, whereas the warm extremes have increased preferentially over northern Canada, southern North America, and a large fraction of Eurasia. What factors are responsible for these surprising spatial and temporal contrasts (pattern correlation between Figs. 1b and 1c is −0.19) in the changes of wintertime cold and summertime warm extremes? For GMST, multiple linear regression (MLR) has proven to be a useful starting point for diagnosing sources of variability. In particular, a linear regression model with four predictors, anthropogenic forcing/linear time trend, the El Niño-Southern Oscillation (ENSO), total solar irradiance (TSI), and volcanic aerosols, is capable of explaining ~70–80% of the global mean temperature variance[17,18]. Given the utility of MLR for diagnosing global mean temperature variability, we construct similar models for NH land wintertime cold and summertime warm extremes for the 1951–2014 period. This analysis is based on the combination of reanalysis and observational data to cover the full period (see Supplementary Note 1 and Supplementary Fig. 2). We first use the same four predictors as in ref. [18] (although TSI is discarded because it did not pass the significance screening; see Methods), but the resulting regressions fail to capture a substantial fraction of the interannual and interdecadal variability (Supplementary Fig. 3). In order to determine the important missing predictors and to improve the regression model, we augment our MLR models through a method called partial least squares regression (PLSR)[19,20]. PLSR determines linear combinations of predictor variables that can improve the regression relationship (Methods). We choose gridded fields of mid-tropospheric (500 hPa) geopotential height (z500) and NH sea surface temperatures (SSTs) as the potential missing predictors to capture the effects of internal atmospheric variability or coupled ocean–atmosphere interactions, which we hypothesize are key missing pieces of our original MLR analysis.

Through this application of PLSR we indeed determine that much of the unexplained variance of land temperature extremes can be captured with an additional z500 predictor, described as the z500 extremes pattern, for wintertime TX10d and an additional SST predictor, described as the SST extremes pattern, for summertime TX90d (Methods). After adding the influence of the z500 and SST extremes patterns, the correlation between the regressed and actual time series improves to 0.89 for TX10d (Fig. 2a) and 0.95 for TX90d (Fig. 2b). Most notably, both regressions capture the increasing wintertime cold and summertime warm extremes during the hiatus. The z500 and SST extremes patterns also emerge as leading patterns in data that exclude the 2002–2014 hiatus period (Supplementary Fig. 4).

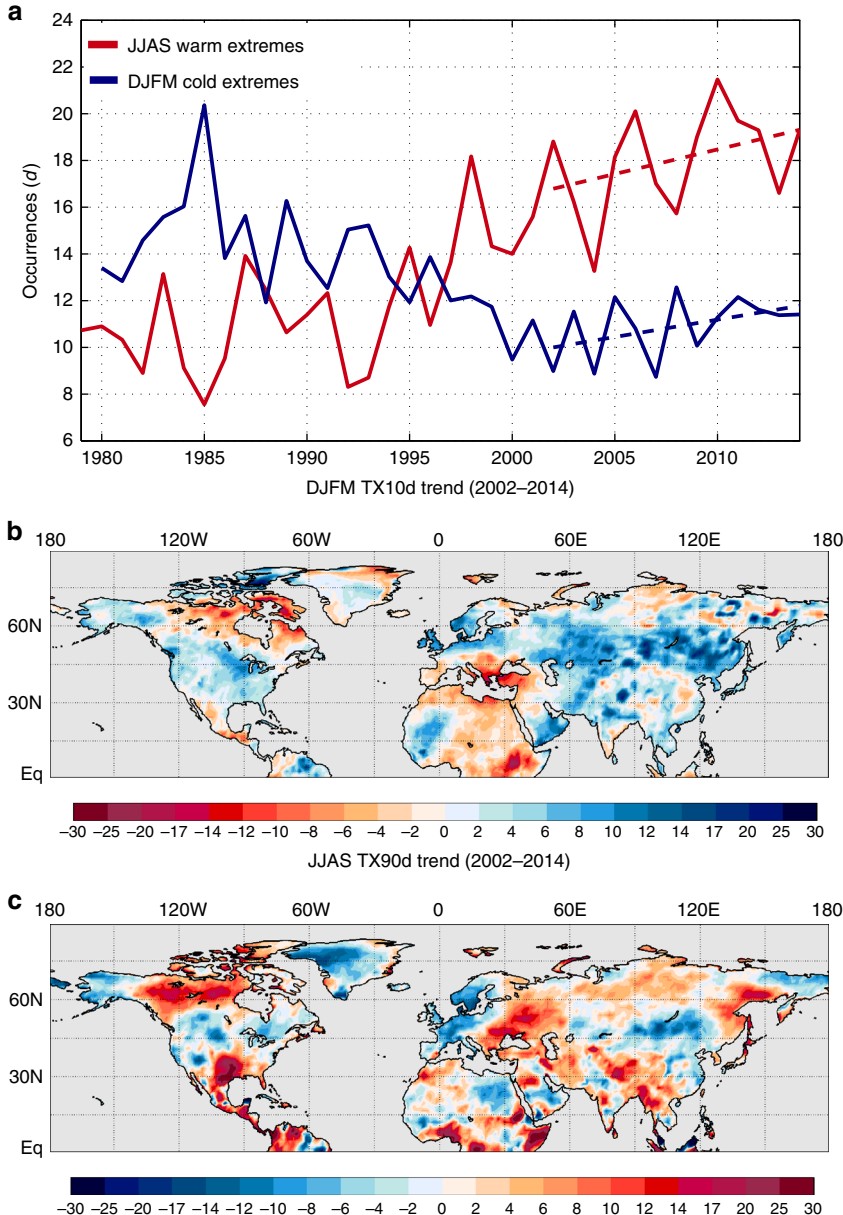

**Fig. 1** Linear trends of wintertime cold and summertime warm extremes during the recent global warming hiatus. **a** Wintertime (December–March, DJFM) cold (blue) and summertime (June–September, JJAS) warm (red) extreme temperature occurrences ($d$ season$^{-1}$) over Northern Hemisphere land from 1979–2014. Temperature extremes are defined by the 10th and 90th percentiles of the local ERA-Interim temperature anomaly distributions (see Methods). Dashed lines indicate the least squares linear trend during the hiatus period of 2002–2014. **b**, **c** Linear trends of wintertime cold extreme temperature occurrence (**b**) and summertime warm extreme temperature occurrence ($d$ [10 yr]$^{-1}$) (**c**) during 2002–2014 at each Northern Hemisphere land grid point

**Contributions to extreme temperature variability**. The time series of individual predictor contributions provides insight into the recent behavior of the temperature extremes. The bottom panels of Fig. 2 are calculated by multiplying the individual predictors (with their time mean removed) by their corresponding regression coefficients at each time step. Major volcanic eruptions resulted in sharp increases in wintertime cold extremes (Fig. 2c) and even sharper decreases in warm extremes (Fig. 2d). Major ENSO episodes, like the extreme El Niño of 1997–98, also were responsible for opposing changes in winter cold and summer warm extremes. During the hiatus period, however, neither volcanic aerosols nor ENSO provided a strong enough influence to offset the overall downward trend of cold extremes or to accelerate the upward trend of warm extremes substantially. This

finding contrasts the dominant role of tropical Pacific SSTs on the global mean warming slowdown[6–8]. Instead, the component time series indicate that the z500 and SST extremes patterns bear most of the responsibility for the anomalous behavior of cold and warm extreme occurrences, respectively, relative to the long-term trend (Fig. 2c, d).

The analysis of the spatial temperature extreme occurrence patterns associated with each predictor further support the dominant roles of the z500 and SST extremes patterns during the hiatus period (see Supplementary Note 2 and Supplementary Figs. 5, 6, and 8). For cold extremes, the z500 extremes pattern (Fig. 3a) features positive mid-tropospheric height anomalies over the high latitudes and negative height anomalies over the midlatitudes, especially over the North Atlantic and Eurasia.

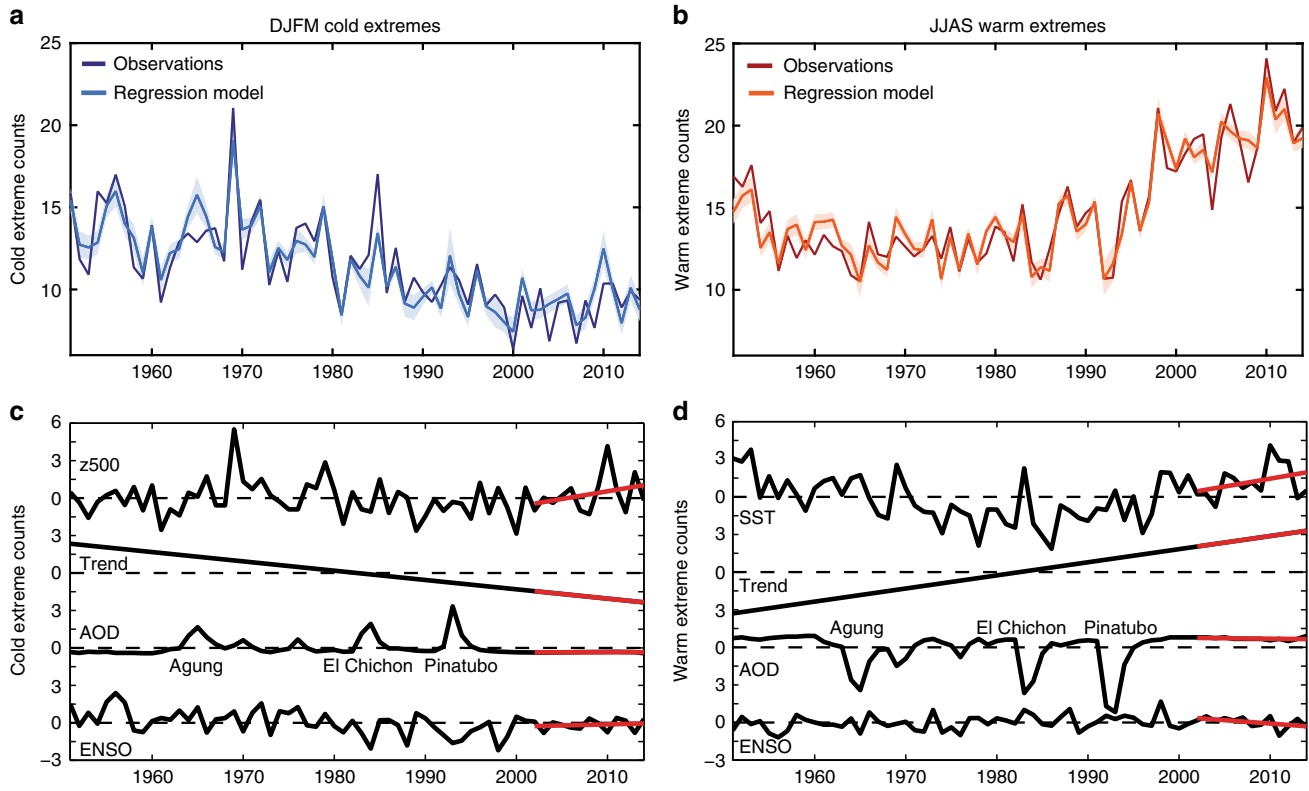

**Fig. 2** Linear regressions of wintertime cold and summertime warm extremes. Time series of wintertime cold (**a**) and summertime warm (**b**) extreme temperature occurrences ($d$ season$^{-1}$) over Northern Hemisphere land from 1951–2014 in observations (dark blue and red) and for linear regression models with four predictors (light blue and orange). Shading indicates the 95% confidence interval for the regression. **c**, **d** Contribution of each individual predictor for the frequency of cold extreme (**c**) and warm extremes (**d**), calculated by multiplying the predictors by their corresponding regression coefficients. Red lines indicate linear trend lines for the hiatus period (2002–2014). Major volcanic eruptions are labeled in the AOD contribution plots. The top regressions in (**c**) and (**d**) indicate the influence of the z500 and SST extremes patterns, respectively

The z500 extremes pattern projects onto the negative phase of the North Atlantic Oscillation (NAO) /Arctic Oscillation (AO). Despite the strong relationship with both the NAO ($r = -0.65$) and AO ($r = -0.68$), the z500 extremes pattern index is much more strongly related to wintertime TX10d than either index (see Supplementary Note 3), indicating that neither the canonical NAO nor AO can explain the recent increase in cold extreme occurrences as well as the z500 extremes pattern. In addition, the z500 extremes pattern appears to be related to the "warm Arctic–cold continents" (WACC) pattern[11], a pattern tied to recent midlatitude extreme weather and wintertime cold temperatures[21–23]. In support of this contention, the z500 extremes pattern is much more strongly correlated with the linearly detrended December–March NH land mean surface temperature ($r = -0.54$) than either the NAO ($r = 0.19$) or AO index ($r = 0.26$) (see Supplementary Note 3). The partial regressions of cold extreme occurrences on the z500 extremes index (Fig. 3e) coincide well with the mid-tropospheric height pattern (Fig. 3a) and bear notable similarities to the linear trend of cold extreme occurrences during the hiatus period (Fig. 1b, Supplementary Fig. 8) and the winter mean temperature trends since 1990[21].

In boreal summer, the SST pattern most closely associated with the recent rapid increase of summertime warm extreme occurrences is associated with a SST pattern with anomalous warmth focused in the North Atlantic (Fig. 3b). The Atlantic portion closely resembles the positive phase of the Atlantic Multidecadal Oscillation[24] (AMO), but significant SST anomalies also occur in the North Pacific. The North Pacific pattern bears a strong resemblance to the Pacific Extreme Pattern[25], a recently identified pattern that was found to provide skillful predictions of hot weather in the eastern U.S. up to 50 days in advance. The SST extremes pattern has exhibited pronounced multidecadal variability consistent with the AMO, with a predominantly positive phase since the mid-1990s (Fig. 3d). Despite a strong relationship with AMO index[26] ($r = 0.66$), the SST extremes index explains much more of the residual summertime TX90d variance (75% versus 35%) and is correlated more strongly with the linearly detrended June–August mean land NH temperature ($r = 0.68$ versus 0.20; see Supplementary Note 3). These calculations reveal that the elements of the SST extremes pattern that are distinct from the AMO are important for hemispheric warm extreme occurrences. The partial regression pattern of warm extreme occurrences on the SST extremes index features pronounced increases in warm extremes of more than three occurrences per season over the southern U.S., Eastern Europe, and southeastern Asia, which bears some similarity to trend pattern of warm extreme occurrences during the hiatus period (Fig. 1c, Supplementary Fig. 8).

The prominence of the Atlantic rather than Pacific SST anomalies is surprising, particularly in light of the importance of Pacific SST variability during hiatus periods[6,7]. Previous work, however, supports that the warm phase of the AMO is associated with positive temperature anomalies in parts of North America and Eurasia during boreal summer[27,28]. In fact, recent climate model studies indicate that the Pacific SST patterns associated with the AMO and the recent (~35-yr) trend may be forced in large part by the Atlantic SSTs through coupled ocean-

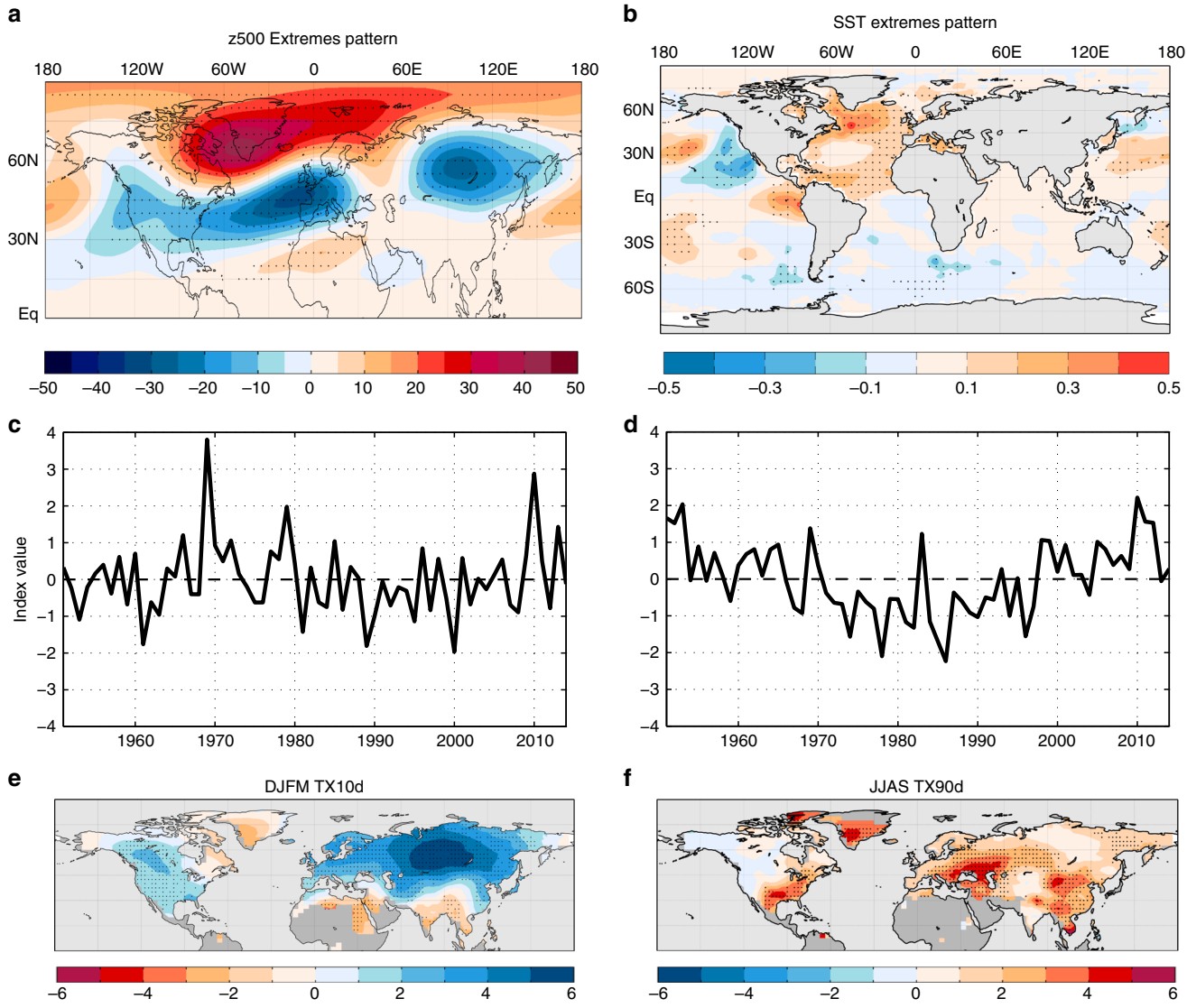

**Fig. 3** The z500 and SST extremes patterns. The wintertime z500 (hPa) (**a**) and summertime SST (°C) (**b**) extremes patterns, where each pattern is a partial regression map representing the change of z500 or SST per standard deviation of the index time series (see Methods). **c**, **d** The standardized index time series corresponding with the z500 (**c**) and SST (**d**) extremes patterns. **e**, **f** Partial regression coefficients of wintertime cold extreme occurrences ($d$ season$^{-1}$) on the z500 extremes index (**e**) and summertime warm extreme occurrences on the SST extremes index (**f**). In the top and bottom panels, stippling indicates regression coefficients that are statistically significant at the 5% level based on a two-sided $t$-test for which the temporal degrees of freedom are adjusted for autocorrelation (see Methods)

atmosphere feedbacks via modulation of the Walker circulation[28–30]. These recent studies suggest that Atlantic Ocean variability may have a stronger impact on global climate than previously recognized.

**Attributing changes in the extremes patterns**. Was the recent emergence of the z500 and SST extremes patterns a manifestation of internal climate variability? As an initial step in addressing this question, we examine the statistical significance of the changes of the observed TX10d and TX90d trends through an analysis of their confidence intervals. We also compare these results with an analysis of a 500-yr simulation of a coupled climate model with high atmospheric resolution, the Geophysical Fluid Dynamics Laboratory (GFDL) Forecast-oriented Low Ocean Resolution (FLOR) model[31], for which radiative forcings are held constant at 1990 values.

First, we examine the variations of the observed TX10d and TX90d linear trends with variable starting years (Fig. 4a). For DJFM TX10d the linear trends do not deviate significantly from the long-term linear trend until the late 1990s; TX10d trends beginning from 1998 to 2004 are significantly larger, as determined by the 95% confidence intervals, than the long-term negative trend of $-4.8$ d (50 yr)$^{-1}$ but not significantly different from zero. The results from the 500-yr climate model simulation support the unusual nature of these recent trends: only 2.9% of the 13-yr TX10d trends from the climate model are larger than the observed 2002–2014 trend with the long-term linear trend removed (dashed blue line of Fig. 4b). This finding is consistent with recent studies suggesting that it would require a particularly extreme realization of internal variability, as determined from state-of-the-art climate models, to explain the recent increase in the occurrence of the WACC pattern and the decreasing wintertime temperatures over Eurasia[23,32–34].

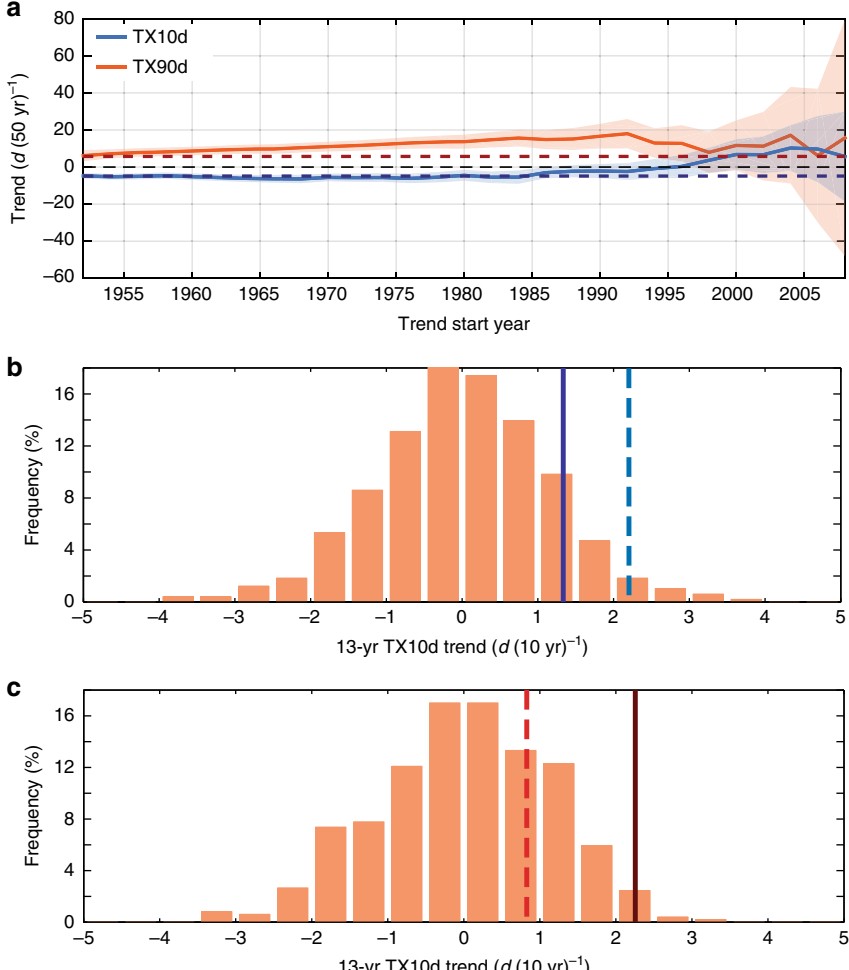

**Fig. 4** Internal variability of Northern Hemisphere wintertime cold and summertime warm extreme occurrences. **a** Linear trends ($d$ [50 yr]$^{-1}$) of DJFM TX10d (blue) and JJAS TX90d (orange) with variable starting years (x-axis) and an end year of 2014. Shading indicates the 95% confidence interval. The dashed blue (red) line indicates the full 1951–2014 linear trend for DJFM TX10d (JJAS TX90d). **b** Histogram of the 13-yr DJFM TX10d trends in the 500-yr FLOR simulation. The dark blue line is the observed value based on HadEX2 combined with ERA-Interim data for the 2002–2014 period. The dashed blue line is the observed value for the 2002–2014 period after the removal of the 1951–2014 linear trend. **c** Same as **b** but for the 13-yr JJAS TX90d trends. The dark red line is the observed value for the period 2002–2014, and the dashed red line is the observed value after the removal of the 1951–2014 linear trend

Another possibility, however, is that such a realization of internal variability is not as extreme as climate models suggest, and that these models underestimate the natural, multidecadal variability of NH wintertime cold extreme occurrences. A recent study[35] indicates that many state-of-the-art climate models underestimate multidecadal NAO variability. Given the strong relationship between the NAO and z500 extremes pattern, it is conceivable that climate models also underestimate the internal variability of NH cold extreme occurrences. Indeed, we find that the distribution of 13-yr DJFM TX10d trends in observations is wider and significantly different from that of the 500-yr FLOR simulation (Supplementary Note 5 and Supplementary Fig. 12). Consistent with this finding and with ref. [35], the FLOR simulation also underestimates the variance of 500 hPa height over the action centers of the z500 extremes pattern (Fig. 5). In the seasonal mean data, the FLOR simulation exhibits more variability in the northeastern Pacific and southern North America (Fig. 5e), a finding that likely relates to the excessive ENSO variability in FLOR[31], given the connection between those regions and strong ENSO episodes (e.g., ref. [36]). Most notably on both timescales, especially in the 13-yr running mean data, the FLOR simulation underestimates the mid-tropospheric height variance over the

North Atlantic and Eurasia (Fig. 5f), two locations that correspond well with action centers in the z500 extremes pattern (Fig. 3a). This analysis indicates that climate models, including high-resolution models like FLOR, may underestimate natural, multidecadal variability of cold extreme occurrences owing to the underestimation of NAO-like variability over the North Atlantic and Eurasia.

An alternative and highly debated hypothesis is that the recent increase of WACC pattern and continental extreme cold occurrences may be caused, at least in part, by increasing greenhouse gases and the resulting Arctic amplification and decrease in sea ice loss[11,21,37–40]. Our findings confirm a statistical link between the z500 extremes pattern and Arctic sea ice, as the z500 extremes pattern index has statistically significant negative partial correlations ($r < -0.4$) with the preceding November Barents-Kara Sea ice anomalies (Fig. 6). However, confidence in the physical connection between Arctic sea ice loss, the WACC pattern, and continental cold extremes is limited by the inability of many climate models to simulate a robust circulation and cooling response to Arctic sea ice loss[23,32,34]. The results presented here cannot refute either hypothesis, and so the connection between Arctic amplification

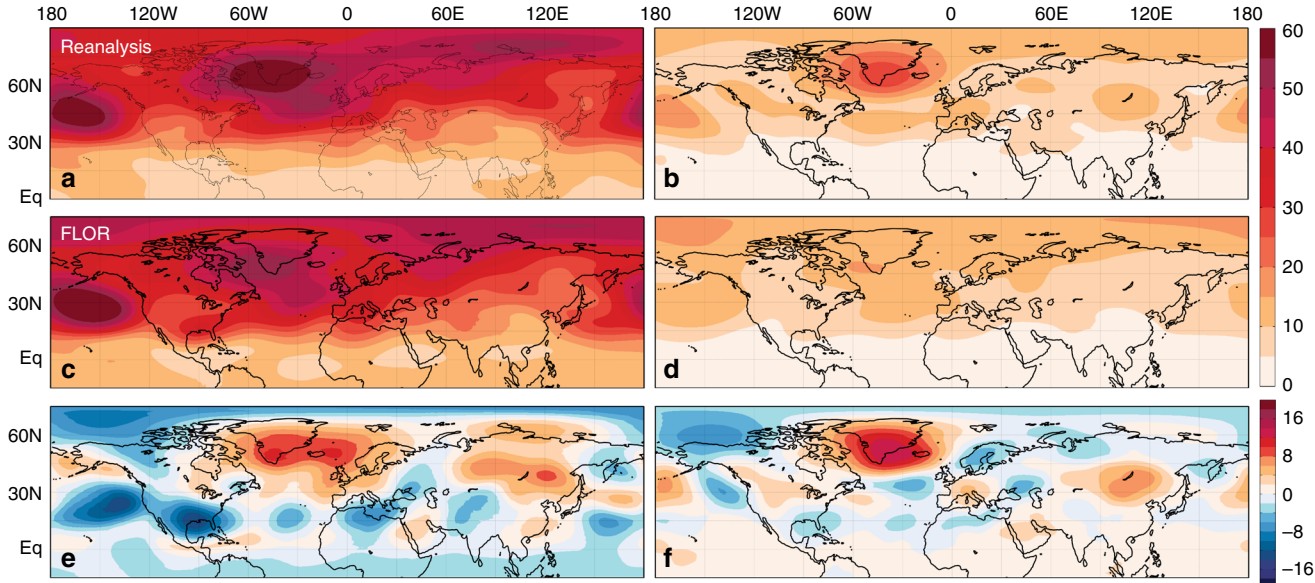

**Fig. 5** Standard deviations of wintertime geopotential height in reanalysis and in FLOR. Standard deviations (*m*) of linearly detrended December–March 500 hPa geopotential height in NCEP/NCAR reanalysis data (**a**) and the 500-yr FLOR simulation (**c**). **b**, **d** Same as **a** and **c** but for 13-yr running mean 500 hPa geopotential height data. **e** Difference between reanalysis and FLOR standard deviations (**a** minus **c**). **f** Same as **e** but for 13-yr running mean data **b** minus **d**

and wintertime cold extremes likely will remain an active area of study (see additional discussion in Supplementary Note 4).

Another hypothesis suggests that the predominance of the negative phase of the NAO, which resulted in frequent cold air outbreaks over Eurasia during the hiatus period, may have been forced, at least in part, by the pattern of tropical Pacific SSTs[9]. This hypothesis, however, contrasts the finding that the negative phase of the NAO typically is connected with the warm phase of ENSO[41], as opposed to the La Niña-like SST pattern during the hiatus period. Indeed, the partial regression of wintertime SST anomalies on the z500 extremes index reveals a weak El Niño-like pattern (Supplementary Fig. 9) that contrasts the tropical Pacific SST anomaly pattern during the hiatus, suggesting that the predominance of the z500 extremes during the hiatus was not the result of tropical SST forcing.

In contrast with the DJFM TX10d linear trends, the JJAS TX90d trends increase gradually as the starting year increases from 1951 until ~1980 (Fig. 4a). This finding indicates a significant acceleration of the positive trend in warm extreme occurrences from the mid to late 20th century, as determined by the failure of the 95% confidence intervals of the linear trends with start years after ~1965 to contain the 1951–2014 linear trend. Therefore, the SST extremes pattern may, in part, capture this nonlinearity in the TX90d trend. This result is consistent with evidence that North Atlantic cooling induced by anthropogenic aerosols projects onto observed Atlantic multidecadal variability[42,43]. However, a large fraction of Atlantic multidecadal variability likely is internally generated, potentially resulting in an apparent acceleration of sea surface warming in the late 20th and early 21st centuries[44]. Consistently, after the effect of ENSO has been linearly removed, the SST extremes pattern emerges in the 500-yr FLOR simulation as a leading contributor to variability in summertime warm temperature extreme occurrence (Supplementary Note 5 and Supplementary Fig. 11). These findings support the existence of the SST extremes pattern as a robust, naturally occurring pattern that can induce apparent accelerations of the warm extreme occurrence trend, as occurred in the 2002–2014 hiatus period. The TX90d linear trends beginning from the mid-1990s are not significantly different from the long-

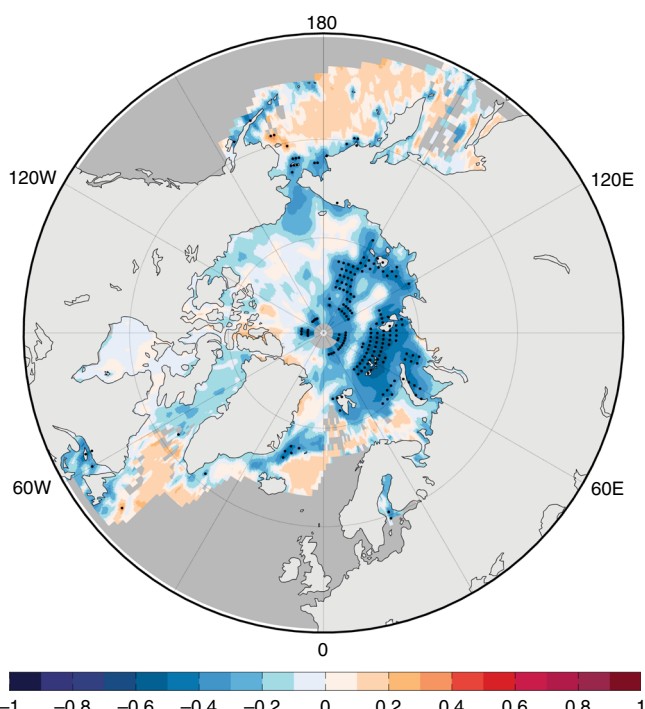

**Fig. 6** Relationship between the z500 extremes index and sea ice. Partial correlation coefficients between the December–March z500 extremes index and preceding November SIC anomalies after linearly removing the influence of ENSO, volcanic AOD, and time trend. Stippling indicates statistically significant correlations at the 5% significance level

term trend (Fig. 4a), indicating that the more rapid increase in warm extremes during the hiatus period associated with the SST extremes pattern is well within the range of internal climate variability. Additional support is found in Fig. 4c: 25.6% of the 13-yr TX90d trends from the FLOR simulation are larger than the observed 2002–2014 trend with the long-term linear trend

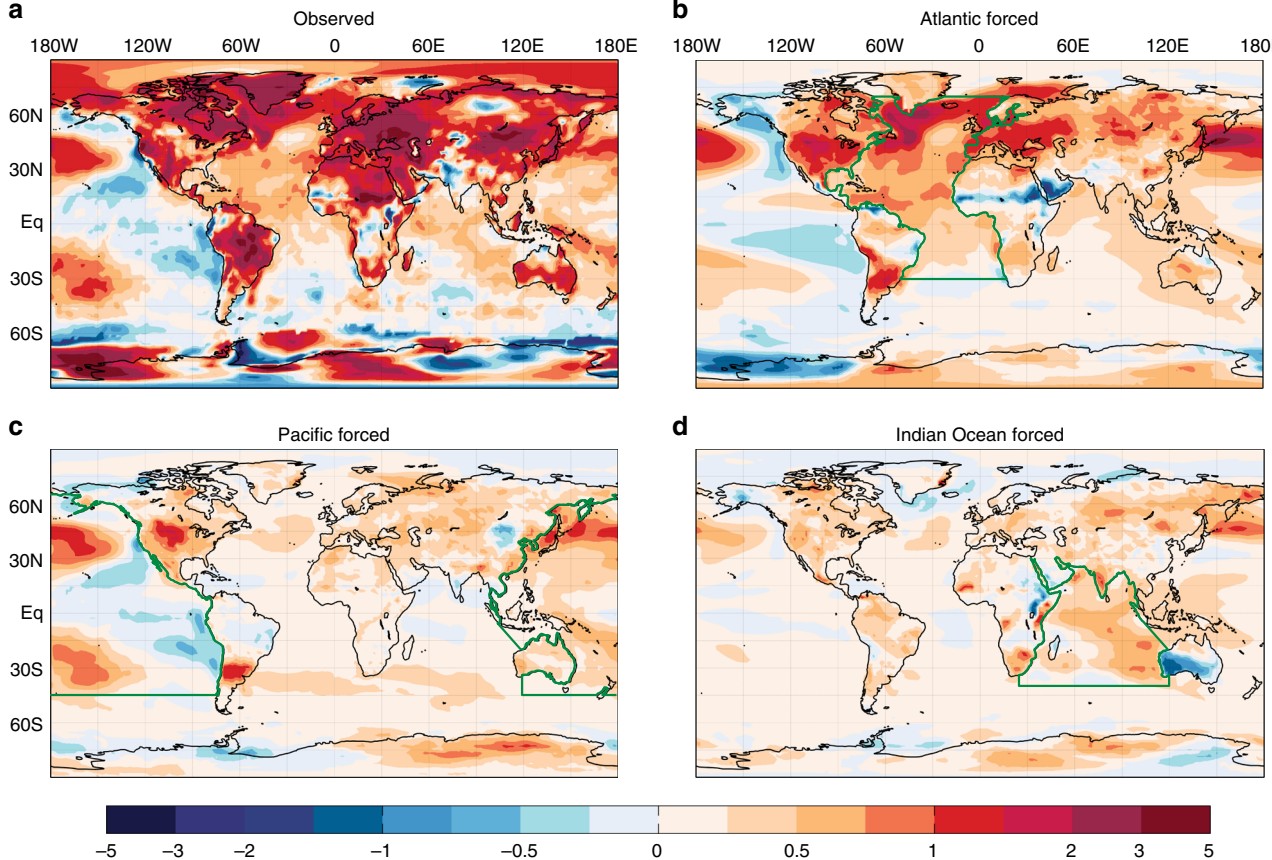

**Fig. 7** Summertime 1979–2012 surface temperature trends forced by ocean basin SSTs. **a** The observed summertime (June–September) 1979–2012 seasonally averaged daily maximum surface temperature trend (°C [34 yr]$^{-1}$), where values over land and regions of sea ice are derived from the ERA-Interim reanalysis and values over all other ocean regions are derived from HadISST. **b**–**d** The surface temperature trend over the same period derived from CESM1 simulations with SSTs nudged to HadISST over the Atlantic (**b**), Pacific (**c**), and Indian Oceans (**d**). The regions outlined in green define the locations of SST nudging (see Methods)

removed (dashed red line), which means that internally driven, apparent accelerations of warm extreme occurrences like what occurred from 2002–2014 are relatively common in the climate model.

Although the analysis presented above confirms a strong statistical link between hemispheric warm extreme occurrences and an AMO-like SST extremes pattern, the analysis does not confirm Atlantic SST anomalies as a dominant cause of the modulation of hemispheric increases in summertime warm extreme occurrences. To solidify a causal connection, we perform simulations with a fully coupled earth system model. Specifically, we simulate the global temperature response to individual ocean basin SST trends with the Community Earth System Model (CESM1, see Methods). In these experiments, the SSTs are nudged to the observed 1979–2012 SST trend pattern, a pattern that resembles both the SST extremes pattern and the May–June 2002–2014 SST anomaly pattern (see Methods), in individual ocean basins (Indian, Pacific, and Atlantic Oceans). Outside the nudging regions, SSTs freely evolve, as in the experiments of ref. [30]. The Atlantic SST nudging experiment results in 63% and 84% more NH land warming than either the Pacific or Indian Ocean SST nudging experiments, respectively (Fig. 7), supporting the dominant role of Atlantic SSTs. Moreover, the pattern of both land and ocean warming outside of the Atlantic basin, including the North Pacific Ocean, bears a strong resemblance to the observed warming pattern (Fig. 7a), which supports the aforementioned studies indicating the importance of the Atlantic Ocean in remote SST trends[28–30]. The stronger warming in the

reanalysis data likely reflects the omission of the direct effects on increasing greenhouse gas concentrations in the SST nudging experiments, which are important for continental warming[45,46].

## Discussion

In summary, global-scale temperature variations were quite unusual from 2002–2014, with relatively little change in global mean temperatures but with contrasting trends in winter and summer extreme temperature occurrence over NH continents. These findings are consistent with the asymmetry in recent seasonal temperature trends noted previously[15], as the arguments presented here apply to seasonal mean temperature in addition to extreme temperature occurrence because of the close correspondence between the two[47,48]. The recent changes in cold and warm extreme occurrences closely track the NH seasonal mean land temperature ($r = -0.86$ for DJFM TX10d and $r = 0.96$ for JJAS TX90d; see Supplementary Fig. 7 and the discussion in Supplementary Note 2). The analysis introduced in this study extends beyond the previous work by highlighting the large-scale climate patterns responsible for such asymmetries and other variations in continental extreme temperature occurrence, which are distinct from those that dominate the variability of annual GMST.

This analysis also underscores the limited usefulness of annual GMST as a measure of the state of the climate owing to the effects of averaging across spatial and time scales. Regional-scale consequences over the NH will not necessarily track annual GMST because NH variability is dominated by different climate modes than those of GMST in different seasons. The results presented

here suggest that for periods when the WACC-like z500 and AMO-like SST extremes patterns remain in a predominantly positive phase, as occurred from 2002–2014, then both winter-time cold and summertime warm extreme occurrences may increase relative to the long-term trends. Whereas GMST is strongly regulated by tropical Pacific SSTs[6–8], variations in the linearly detrended NH land mean surface temperature are strongly related to the z500 extremes pattern in DJFM ($r = -0.54$) and to the SST extremes pattern in JJAS ($r = 0.68$).

In this study, we have used a lenient definition of extremes by focusing on occurrences below the 10th and above the 90th percentiles of the temperature anomaly distributions. This choice allows us to analyze relatively large samples of extreme temperature occurrences, thereby yielding robust statistics. A limitation of this approach, however, is that we do not address changes in the most 'extreme' hot extremes, which have risen more dramatically than less extreme heat events during the hiatus period[10]. Indeed, we find that the most severe summertime NH hot extremes have risen faster than more moderate hot extremes during the hiatus period, although more severe cold extreme occurrences have not experienced a similar amplification (Supplementary Note 6 and Supplementary Fig. 13). We suspect that the dramatic rise of the most extreme heat waves involves local-scale interactions such as land–atmosphere feedbacks, which play an important role in the projected increase of hot extremes[49,50], and possibly land use and land cover changes[51] in addition to favorable large-scale atmospheric and oceanic conditions. More work is needed to understand how increasing greenhouse gases, large-scale modes of climate variability, and local-scale feedbacks will interact to alter the characteristics of temperature extremes in a warming world.

## Methods

**Temperature extreme counts**. For the analysis focusing on the satellite era (1979–2014), we use six-hourly ERA-Interim reanalysis[52] and follow the procedure described by the Expert Team on Climate Change Detection and Indices (ETCCDI) (http://www.clivar.org/organization/etccdi) to calculate the number of cold and warm extreme days in December–March (DJFM) and June–August (JJAS), respectively. Specifically, we identify the daily maximum temperature ($T_{max}$) at each land grid point from 1979–2014 and then remove the 1981–2010 seasonal cycle to define daily $T_{max}$ anomalies. A cold extreme on day $d$ is identified if the $T_{max}$ anomaly falls below the 10th percentile of the $T_{max}$ anomaly distribution defined by all 5-day intervals from 1981–2010 centered on the calendar day. Similarly, a warm extreme is identified if the $T_{max}$ anomaly lies above the 90th percentile of the $T_{max}$ anomaly distribution. To avoid possible inhomogeneities across the in-base and out-of-base periods, we follow the bootstrap procedure described in ref. [53]. We then count the number of cold extreme occurrences (TX10d) in each DJFM (February 29 is excluded) and the number of warm extreme occurrences (TX90d) in each JJAS.

For the multiple linear regression (MLR) analysis described below, we also use the TX10d and TX90d data from the HadEX2 dataset[54] for the period of 1951–2010. The HadEX2 dataset is derived from high-quality in situ observations from meteorological stations around the world, interpolated to a 2.5° x 3.75° latitude–longitude grid. In addition to a basis in quality-controlled observations, the HadEX2 data have the advantage of a longer record than the ERA-Interim data, allowing more robust statistical analyses. Disadvantages include incomplete spatial coverage and a termination in 2010, which excludes the later years of the hiatus period. To cover the full hiatus period of interest, we extend the HadEX2 data to 2014 through the following linear regression relationship:

$$T_{H2} = \beta_1 T_{EI} + \beta_0 \qquad (1)$$

where $T_{H2}$ is the HadEX2 extreme temperature measure, either a grid point value or the area mean, depending on the application, and $T_{EI}$ is the corresponding ERA-Interim value. The regression model is trained for the 1979–2010 period of overlap, and the extrapolation occurs for 2011–2014. The ERA-Interim data are linearly interpolated to the HadEX2 grid, and for the area mean calculations, only grid points for which HadEX2 has at least 90% temporal coverage are considered (see Fig. 3e, f for an indication of spatial coverage). For the period of overlap, the correlation between the Northern Hemisphere areal mean HadEX2 and ERA-Interim temperature extremes is 0.98 for DJFM TX10d and 0.99 for JJAS TX90d, which lends credibility to the regression procedure and the reliability of both datasets. The grid point correlations, shown in Supplementary Fig. 2, are generally

higher than 0.8. The intercept term $\beta_0$ corrects for the offset in base periods (1981–2010 for ERA-Interim and 1961–1990 for HadEX2).

**Choice of hiatus period**. We define the hiatus period as 2002–2014 following ref. [7] but extending the end year from 2012 to 2014 to include more data before the extreme El Niño of 2015/16. The general conclusions, however, are not sensitive to the choice of hiatus period. For example, if we define the hiatus period as 1998–2012, as in several other studies (e.g., ref. [9]), the linear trends in DJFM TX10d, JJAS TX90d, z500 extremes index, and SST extremes index maintain the same sign as in the 2002–2014 period.

**Multiple linear regression**. We build multiple linear regression (MLR) models of the form

$$T_{comb} = \beta_y y + \beta_E E + \beta_A A + \beta_0 + \varepsilon \qquad (2)$$

where $T_{comb}$ is the NH land mean extreme count from the combined HadEX2 and ERA-Interim data, covering the period from 1951–2014, $y$ is the year, $E$ is the ENSO index, $A$ is the volcanic aerosol optical depth (AOD), $\beta_i$ is the regression coefficient for predictor $i$ ($\beta_0$ is the intercept term), and $\varepsilon$ is the residual. We also consider total solar irradiance (TSI) as a potential predictor, but TSI did not pass the significance screening described below. For the choice of ENSO index, we considered both the Niño 3.4 SST index and the Multivariate ENSO Index (MEI)[55]. The correlation between $T_{comb}$ and $E$ generally is higher for MEI, particularly for cold extremes, so we retained the MEI as the ENSO predictor. We use AOD data from ref. [56]. The AOD and MEI data are of monthly and bi-monthly temporal resolution, respectively. The TSI data that we tested were from refs. [57,58], where the data were extended over the full period following a regression approach as in Eq. (1), and correlations are well over 0.9 during the period of overlap.

Given the apparent nonlinearity in the TX90d trends during the period of interest (Figs. 2 and 4), one may consider using anthropogenic forcing instead of time as a predictor. We choose to use time as a predictor for two main reasons. First, a measure of anthropogenic forcing requires an estimate of anthropogenic aerosol forcing, which has substantial uncertainty over the period of interest. Second, the regression models as currently constructed perform exceptionally well (Fig. 2), which indicates that the z500 and especially SST extremes patterns, described below, can capture any nonlinearity in the TX10d and TX90d trends. Additional synthesis allows us to further diagnose how both internal variability and radiative forcing may contribute to their variations and therefore to the nonlinearity in the TX10d and TX90d trends. Consequently, the current analysis allows us to highlight how both radiative forcing and internal variability may have similar spatial fingerprints with respect to dominant predictors of Northern Hemisphere temperature extremes.

Following previous conventions[17,18], we determine the predictor lags by the maximum correlation between the predictor and predictand (except for the trend term). The lags for DJFM TX10d are −7 months (June) for both the MEI and AOD (note that the negative sign indicates that the predictor leads the predictand). The lags for JJAS TX90d are −11 months (August) for the MEI and −10 months (September) for the AOD. These lags, particularly for the MEI, are longer than those that maximize the relationship with global mean temperature[17,18]. A possible explanation is that stronger ENSO events, which tend to peak in boreal autumn but also reach a mature stage earlier in the boreal summer than the weaker events, may have a disproportionately strong influence on temperature extremes. Also, the peak correlations for AOD are not particularly pronounced. As a result, we tested several versions of the MLR model, varying the lags of the predictors and substituting the Niño 3.4 SST index for the MEI, and all results were similar for each model. We calculated the correlations between each pair of predictor variables, and all correlations were at or below 0.40 for DJFM TX10d (strongest correlation of 0.40 between MEI and time) and 0.20 for JJAS TX90d (strongest correlation of 0.19 between MEI and volcanic AOD). Therefore, the predictor variables are not nearly collinear. The final models are chosen to ensure that all predictors are significant at the 10% level, where the significance of each predictor is assessed with a partial F-test. All predictors except for MEI in the TX90d regressions ($p = 0.056$) have p-values well below 0.05. We also visually inspected normal probability plots to ensure that the residuals are approximately Gaussian. The 95% confidence interval of the fitted value for time $t$ is given by

$$\hat{T}_{y,CI} = \hat{T}_y \pm t_{0.025}\sqrt{\hat{\sigma}^2 \mathbf{x}'_y (\mathbf{X}'\mathbf{X})^{-1} \mathbf{x}_y} \qquad (3)$$

where $\hat{T}_y$ is the regressed value of $T$ in year $y$, $\hat{\sigma}^2$ is the mean square error, $\mathbf{X}$ is the design matrix, $\mathbf{x}_y$ is the predictor values in year $y$ in column vector form, and all other notation is standard. These linear regressions are illustrated in Supplementary Fig. 3.

**Partial least squares regression**. In order to improve the regression models given by Eq. (2), we seek additional predictors through the method of partial least squares regression (PLSR)[19,20]. We consider NH 500 hPa geopotential height (z500) from NCEP/NCAR reanalysis[59] and ERSSTv3b[60] SST from 20°S to 60°N from lags of −12 to 0 months as potential predictors of DJFM TX10d and JJAS TX90d. We first calculate three-month (four-month for lag 0 exceptionally) seasonal mean

standardized anomalies of z500 and SST by subtracting the seasonal cycle and dividing each anomaly by the grid point standard deviation. Then we follow the general procedure for PLSR that is discussed in ref. [20]: (1) linearly remove the three predictors in Eq. (2) from both the areal mean temperature extreme counts and the gridded z500 and SST anomalies to determine residual predictand (TX10d or TX90d) and predictor fields; (2) calculate the correlation map between the residual predictand and gridded predictors to determine the PLS predictor pattern; (3) project the residual gridded predictors onto the PLS predictor pattern to determine a PLS predictor time series; and (4) incorporate the PLS predictor time series into regression Eq. (2) as the fourth predictor. It is possible to repeat these four steps to remove the preceding PLS predictors and add additional PLS predictors, but we choose to consider only one potential PLS predictor for the sake of keeping our model as simple and interpretable as possible and because this methodology is prone to overfitting when multiple PLS components are retained. Essentially, this methodology seeks linear combinations of z500 or SST that can be used to generate a predictor time series that explains the maximum amount of variance in TX10d or TX90d unaccounted for by the original three predictors.

After performing the iterative procedure for both SST and z500 and for all lags, we find that the lag 0 (DJFM) z500 PLS predictor for TX10d and lag -1 (MJJ) SST predictor for TX90d emerge as the additional predictors that are capable of explaining most of the missing hemispheric mean temperature extreme variance. The z500 predictor explains 27.0% of the TX10d variance and the SST predictor explains 31.4% of the TX90d variance. The z500 and SST index time series are defined as the standardized projection time series determined in step 3 above (Fig. 3c, d). The z500 and SST predictor patterns, which we call the z500 and SST extremes patterns, are determined by the partial regressions (i.e., ENSO, AOD, and time trend are first linearly removed) of z500 and SST anomalies on the corresponding index time series (Fig. 3a, b).

To evaluate the robustness of the z500 and SST extremes analysis, we repeated the calculations described above but after partitioning the data into a 1951–2001 training set and a 2002–2014 validation set. The purpose of this analysis is to determine if the z500 and SST extremes patterns emerge in data that exclude the hiatus period and if their predicted relationships with extreme temperature occurrence during the hiatus period are consistent with the relationships revealed in Figs. 2 and 3. For these calculations, we removed the influence of the time trend, ENSO, and volcanic AOD from the 2002–2014 DJFM TX10d, JJAS TX90d, the gridded SST, and the gridded z500 fields, just as in the original analysis, but in this analysis all regression coefficients were determined from the 1951–2001 training period. We then performed PLSR analysis to determine the z500 extremes pattern, SST extremes pattern, and regression coefficients of the residual TX10d and TX90d onto the corresponding extremes pattern index with the 1951–2001 data. Finally, we predicted the z500 and SST extremes pattern contributions to TX10d and TX90d, respectively, during the 2002–2014 period. We then compared these out-of-sample calculations with the in-sample calculations reported in Figs. 2 and 3.

Overall, we find that the z500 and SST extremes patterns are robust, confirming that the patterns and their relationships with extreme temperature occurrence were not unique to the hiatus period. The z500 and SST extremes patterns determined from the 1951–2001 training set, shown in Supplementary Fig. 4, are very similar to the patterns shown in Fig. 3. In addition, Supplementary Fig. 4 illustrates the out-of-sample predictions of the z500 and SST extremes pattern contributions to DJFM TX10d and JJAS TX90d, respectively, in comparison with the in-sample partial regressions reported in Fig. 2. Differences between the in-sample partial regressions and out-of-sample predictions may relate to the following: (1) differences in the ENSO, time trend, and AOD regression coefficients that affect the adjustment of the TX10d and TX90d time series prior to the PLSR analysis; (2) differences in the z500 and SST extremes patterns between the two different datasets, and (3) differences in the z500 and SST extremes pattern regression coefficients, holding the z500 and SST patterns identical for the two datasets. Overall, we see some differences between the in-sample and out-of-sample time series, but the hiatus period trends and much of the interannual variability are in good agreement.

The actual and regressed TX10d and TX90d time series with all four predictors are shown in the top of Fig. 2. The 95% confidence interval is again calculated with Eq. (3). We note that this confidence interval does not account for the uncertainty in the PLS predictor patterns themselves, and so the uncertainty likely is underestimated somewhat; however, all conclusions are unlikely to be affected. The time series of individual predictor contributions (bottom of Fig. 2) are calculated by subtracting the time mean of each predictor and multiplying the residual by the corresponding regression coefficients at each time step. Therefore, the sums of the individual contributions in the bottom of Fig. 2 are equal to the full regressions in the top of Fig. 2 with their time means removed. In the partial regression maps (Fig. 3), statistical significance of the regression coefficients is assessed with a two-sided t-test, and degrees of freedom are corrected for autoregression in the residuals following ref. [61].

**Confidence intervals of the linear trends**. For the calculations illustrated in Fig. 4a, we use ordinary least squares regression to calculate the DJFM TX10d and JJAS TX90d linear trends for starting years varying from 1952 to 2008 and ending in 2014. We also calculated the trends with the Theil-Sen method, but the results were very similar to those of simple linear regression. To calculate the 95% confidence intervals, we sought an appropriate model for the noise about the linear fit

to the data by examining the sample autocorrelation of the residuals. For TX10d, the autocorrelation of the residuals is quite small (lag-1 autocorrelation = −0.03), and so we use the standard white noise model to calculate the confidence intervals. For TX90d, the sample autocorrelation is much larger and more persistent than we would expect from a first-order autoregressive (red noise) process. This behavior is quite similar to that of GMST[18]. Consequently, we follow the procedure of ref. [18] and model the TX90d residuals with an autoregressive moving average (ARMA (1,1,)) model. For these calculations, the standard error of the estimated trend is inflated relative to that of the white noise estimate to account for the reduced degrees of freedom. Specifically,

$$\sigma_c = \sigma_w \sqrt{\nu} \qquad (4)$$

where $\sigma_c$ is the corrected standard error, $\sigma_w$ is the white noise estimate of the standard error, and $\nu$ is the inflation factor. For the ARMA(1,1) model

$$\nu = 1 + \frac{2\rho_1}{1 - \varphi} \qquad (5)$$

where $\rho_1$ is the lag-1 autocorrelation coefficient, and $\varphi$ is the autocorrelation decay rate, estimated by

$$\varphi = \frac{\rho_2}{\rho_1} \qquad (6)$$

**FLOR simulation**. To place the recent temperature extremes trends in a broader context, we compare the observed 2002–2014 TX10d and TX90d trends to those of a long control simulation of a coupled climate model. We use a 500-yr simulation of the GFDL-FLOR model[31,62], for which radiative forcings are held constant at 1990 values. The FLOR model features high horizontal resolution in its atmosphere and land components (~50 km) but retains the lower resolution (~100 km) of the GFDL Coupled Model version 2.1 in its ocean and sea ice components. FLOR has demonstrated success in various applications of seasonal climate forecast[31,63–65] and climate change[66] studies.

For the 500-yr simulation, we calculate the DJFM TX10d and JJAS TX90d from daily data in the same way as for the observational data, but here the percentiles are calculated with respect to the full 500-yr period. Despite that radiative forcings are held constant, the model undergoes a gradual warming drift, and so the extreme temperature occurrences are linearly detrended prior to the analysis. The TX10d and TX90d time series exhibit substantial decadal to multidecadal variability despite no variation in the radiative forcing (Supplementary Figs. 10a and 11a), although the decadal TX10d variability is underrepresented in the FLOR simulation (see the main text and Supplementary Note 5 for more discussion). For comparison with the observed TX10d and TX90d trends during the hiatus period, we generate histograms of all 13-yr trends in the simulation (Figs. 4b, c).

**Relationship with Arctic sea ice**. In order to examine the possible relationship between the z500 extremes pattern and Arctic sea ice anomalies in the preceding autumn, we calculate partial correlations between the z500 extremes index and Arctic sea ice concentration (SIC) anomalies in the preceding November. We use monthly SIC data obtained from the National Snow and Ice Data Center (NSIDC) derived from brightness temperature measured by satellite using the NASA Team algorithm[67]. We calculate monthly anomalies by subtracting the 1981–2010 seasonal cycle. Prior to calculating the correlations, we linearly regress out ENSO, volcanic AOD, and the time trend from the SIC anomalies, just as in the z500 anomalies prior to calculating the z500 extremes pattern except that the SIC anomalies only cover the period from 1979–2013.

Figure 6 illustrates the partial correlations between the wintertime (December–March) z500 extremes index and the preceding November SIC anomalies. For the statistical significance calculations, the degrees of freedom were adjusted following the lag-1 autocorrelation adjustment described in ref. [61]. Similar correlation patterns are obtained if we consider the preceding autumn (September–November) or summer (June–August) SIC anomaly fields (not shown), although the area with statistically significant correlations decreases with lag. Because the time trend is one of the predictors that is removed prior to the calculations, these relationships are not related to the long-term downward linear trend of Arctic sea ice.

**Correlations with seasonal mean NH land temperature**. The seasonal mean linearly detrended Northern Hemisphere land surface temperature (NHLST) data used in the reported correlations with the z500 and SST extremes indices are based on land surface temperature for the period 1951–2014 from the Berkeley Earth Surface Temperature project[68], which are monthly land surface temperature data on a 1° latitude–longitude grid. After linearly removing the influence of ENSO, volcanic AOD, and linear time trend, the partial correlation between the SST extremes index and the JJAS NHLST rises to 0.84, and the partial correlation between the z500 extremes index and the DJFM NHLST falls to −0.71.

**CESM experiments**. The National Center for Atmospheric Research (NCAR) coupled climate model, the Community Earth System Model (CESM1.06)[69] was used to investigate the role of observed basin-scale SST trends on the amplitude and pattern of summertime daily maximum temperature trends over the 1979–2012 period. We note that the annual mean 1979–2012 SST trend bears a close resemblance to both the SST extremes pattern (pattern correlation = 0.55) and the 2002–2014 May–June SST anomaly pattern (pattern correlation = 0.74). This analysis focuses on the influences of ocean temperature trends in three basins: Atlantic, Indian, and Pacific basins. The boundaries of each basin are depicted in Fig. 7, with a 10° buffer equatorward of the northern and southern boundaries over which the SST restoring, described below, ramps up from zero. The atmospheric component of this model is the Community Atmospheric Model version 4 (CAM4[70]) with F19 horizontal resolution (~2°). The oceanic component is the Parallel Ocean Program version 2 (POP2), with ~1° horizontal resolution. The basic experimental framework of this analysis is similar to that of ref. [30]. We restored basin-scale ocean mixed-layer temperature in the coupled model as follows:

$$F = cD(T_{v} - T_{m})/\tau \qquad (7)$$

where $c$ is the heat content of sea water, $D$ is the mixed layer depth, $T_{v}$ is the restoring target temperature, $T_{m}$ is the model temperature at each time step, and $\tau$ is the restoring time scale, which was set as 10 days in this study. In these perturbed simulations, we added external heating $F$ to the model to restore basin-scale ocean mixed-layer temperature. The restoring target temperatures define control and perturbed simulations. The climate response to the basin temperature trends was calculated by the difference between a control run and the perturbed run that is restored to the observed ocean trend. In the control run, the mixed-layer temperature was restored to the model climatology. In the perturbed run, the observed temperature trend for 1979–2012 for the particular basin was added to the mixed-layer temperature restored in the control run. The observed trend is derived from the Hadley Centre Sea Ice and Sea Surface Temperature data set (HadISST)[71]. We conducted 12-member ensemble simulations with different initial conditions in the control and perturbed runs. The model was integrated for 15 years and the last 10 years were used for analyses.

**Code availability**. The data in this study are analyzed with widely available tools in Matlab. Contact N.C.J. for specific code requests.

**Data availability**. The observational data that support the findings are publicly available. ERA-Interim data are available at http://apps.ecmwf.int/datasets/data/interim-full-daily/levtype = sfc/. HadEX2 and HadISST data are available at the Met Office Hadley Centre website (https://www.metoffice.gov.uk/hadobs). NCEP-NCAR reanalysis, ERSST, and MEI data can be found at the NOAA/OAR/ESRL PSD website (http://www.esrl.noaa.gov/psd). Berkeley surface air temperature data are available at the Berkeley Earth website (http://berkeleyearth.org/). The AOD, TSI, and AMO time series are available at the KNMI Climate Explorer website (https://climexp.knmi.nl/). The NAO and AO index time series are found at the NOAA Climate Prediction Center website (http://www.cpc.ncep.noaa.gov/products/precip/CWlink/daily_ao_index/teleconnections.shtml).

**Model availability**. Contact N.C.J. for FLOR simulation data requests. Contact N.C.J. or X.L. for CESM simulation data requests.

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

## Acknowledgements

We thank Drs. Hiroyuki Murakami and Dawei Li for their helpful comments on an earlier version of the manuscript. N.C.J acknowledges support from award NA14OAR4320106 from the National Oceanic and Atmospheric Administration, U.S. Department of Commerce. S.P.X. and X.L. were supported by the National Key R&D Program of China (2016YFA0601800) and U.S. National Science Foundation (1637450). Y.K. was supported by Japan Society for the Promotion of Science (Grant Number 15H05466), Japan Science and Technology Agency through Belmont Forum CRA "InterDec", and the Japan Ministry of Education, Culture, Sports, Science and Technology through "Integrated Research Program for Advancing Climate Models" and "Arctic Challenge for Sustainability" projects. NCEP-NCAR reanalysis and ERSST data were provided by the NOAA/OAR/ESRL PSD, Boulder, Colorado, USA, from their Web site at http://www.esrl.noaa.gov/psd.

## Author contributions

N.C.J. led this work with contributions from all authors. N.C.J. performed the analyses of observational and climate model simulation data and led the writing of the paper. S.P.X. contributed to the central ideas of the study. Y.K. assisted with the sea ice data and figures. X.L. ran the CESM simulations. All authors contributed to the analysis design, interpretation of results, and writing of the paper.

## Additional information

**Competing interests:** The authors declare no competing interests.

