## [Peer Review File · Nature Communications]

Reviewers' comments:

Reviewer #1 (Remarks to the Author):

This study diagnoses changes in winter and summer temperature extremes in the Northern Hemisphere during the so-called hiatus period which the authors define from 2002-2014. The authors find that both the number of cold extremes during winter and warm extremes during summer have increased during this period. These trends are different from expectations from changes in global mean temperatures which did not exhibit a change during the hiatus period. The authors utilized a multiple linear regression model. This model was improved by including additional relevant predictors through the method of partial least squares regression. They find that most relevant for the trends in occurrence of temperature extremes during winter is a circulation pattern that resembles the "Warm Arctic-Cold Continent" pattern. They also find that during summer a sea surface temperature pattern that resembles the Atlantic Multi-Decadal Oscillation results in the increase in summer temperature extremes occurrence.

I cannot recommend this manuscript for publication because of the following main concerns.

1. The current study does not add any new fundamental insight into the current understanding of the regional nature of temperature change over Northern Hemisphere land during the hiatus period.
 - 1a. Specifically, the role of the NAO during the hiatus period is described before (e.g. Trenberth et al. 2014, Nature Climate Change). The Trenberth et al. study highlights the role of the NAO during the cold winter of 2009-2010, 2010-2011 and 2012-2013 at the end of the hiatus period that are contributing to a negative cooling trend over Eurasia and Eastern United States. The Trenberth et al. study also suggests a mechanism of tropical Pacific origin that affects NAO/cold Eurasian temperatures and Arctic warming. Furthermore, the Trenberth et al. study also highlights the role of the PDO affecting specifically the cooling signal over western United States. Taking into account that the regression model outcome applied in the submitted study is very similar for seasonal means and extremes occurrence, it is not clear what fundamental new findings are provided.
 - 1b. The role of AMO in the hiatus has been discussed before (See Medhug et al. 2017 and references therein). The authors of the submitted study do not reconcile their findings with the findings in the literature on this topic. Furthermore, the described results of CESM1 experiments showing trends for the period 1979-2012 do not explain the increased hot extremes during the period 2002-2014. How has the AMO changed during that period superposed on the anthropogenic signal?
2. Definition of the hiatus period: The authors of this submitted study define the hiatus period from 2002-2014. This period strongly differs from definitions in the literatures using the period 1998-2012 or 1998-2013 (e.g., Medhug et al. 2017, Nature; Trenberth et al. 2014, Nature Climate Change). The reasoning for using the period 2002-2014 is not given.
3. Utilizing of the term "Warm-Arctic -Cold Continent" (WACC) circulation pattern: The authors do not provide any evidence for the occurrence and importance of such a prescriptive term for the circulation pattern instead of referring to the NAO which resembles their identified 500hPa extremes circulation pattern (as the authors point out themselves). The NAO which is a well-defined mode of variability varies strongly on decadal to multi-decadal time scale with well-established effects on the climate on Northern Hemisphere continental climate. Furthermore, it becomes obvious that recent negative NAO events that occurred during the second half of the studied period contributed to their result. Finally, the physical mechanism of such a WACC pattern that is different from the NAO itself is not clear. Other recent studies point to the impact of a different circulation phenomenon, namely the strengthening of the Siberian high as a main driver of the Eurasian cooling trend over the 1990 to 2014 period when the so-called WACC-like temperature trend pattern occurred (Sun et al. 2016, GRL).
4. As a whole, investigating trends in temperature extreme occurrences averaged over the Northern

Hemisphere land instead of seasonal mean temperature trends does not provide any additional insight into regional land temperature changes during the hiatus period specially over such a very short record. Specifically, these short time series (13 years) of extreme occurrence are very noisy and regression lines to illustrate trends are strongly affected by single years.

Reviewer #2 (Remarks to the Author):

The authors investigate temperature extremes during the recent so-called 'hiatus' period when global mean surface temperature did not change significantly. They show that during this period, however, summertime hot extremes continued to increase on Northern Hemisphere. And at the same time also wintertime cold extremes did increase in frequency. They establish statistical relationships with potential drivers and find that the increase in winter cold extremes is strongly related to a specific circulation pattern in the mid-troposphere, whereas the increase in summer hot extremes seems related mainly to SST patterns similar to AMO. They also find some of these relationships confirmed in climate model simulation.

The manuscript is very well and clearly written, and addresses a scientific gap as changes in temperature extremes during the 'hiatus' period have previously been reported but not explained. I think this is a very nice and comprehensive study to explain these potential drivers of decadal-scale changes in temperature extremes. A few points, however, should be clarified:

- Also the annual global land average temperatures continued to increase during the 'hiatus' period (see e.g. Seneviratne et al 2014, doi:10.1038/nclimate2145). So how do these increases in summer hot extreme temperatures differ from the increasing average temperatures over land?

- Cohen et al. 2012 (doi: 10.1029/2011GL050582) documented seasonal asymmetries in temperature change (incl. summer vs winter). The authors included a brief discussion around differences between seasonal means and seasonal extremes, but I think it could be clarified in how far these 'extreme' changes relate or possibly exceed the mean changes. Given the authors use measures of relatively moderate extremes (that occur on average on 10% of the days), I expect the results between these extremes and the seasonal averages to be reasonably similar. However, there may be larger differences when looking at some more extreme measures of hot/cold extremes.

- As a conclusion, the authors may want to highlight that GMST probably is not a particularly useful measure for relevant climate states or of climate change in general, as it may average out different kinds of events, seasonal and regional characteristics

A few specific comments:

Line 7: it should be specified if the increase is e.g. in frequency or intensity or associated temperature. In particular for cold extremes the term "increase" can be ambiguous: increasing frequency would be consistent with cooler conditions, increase in associated temperatures with warming.

Line 93/94: do you really mean "anomalous warming", or rather "anomalously warm" (i.e. displaying a warm anomaly but not necessarily intensifying)?

Line 116: please specify how you "examine the significance"

Line 117: specify that "high resolution" refers to the atmospheric model, otherwise it seems in contradiction to "Low Resolution" in the following line

Line 125: not clear what exactly "this conclusion" refers to

Line 144: it would be preferable to reserve the use of "significant" when you mean "statistically significant" (and then also explicitly specify the statistic meaning in those occurrences).

Line 182: better "levelled" than "level"?

Line 186: "occurrence of...occurrences" – better reword?

Line 190/191: The causality is not clear: it seems like you are saying the (relatively small) size of NH land area is responsible for its larger variability? It should also be clarified "larger" than what?

Line 417: was the p-value / degrees of freedom corrected for auto-correlation in the fields (see Wilks 2016, <https://doi.org/10.1175/BAMS-D-15-00267.1>)?

Line 436: "surface temperature" – specify it is the average temperature

Line 448: The "satellite era" continues until present, so it would be good to specify why the analysis ends in 2014. Presumably because GMST increased after this?

Comments on the Supplementary information

Title: the title is different to the title of the main text

S2: Please specify if also the merged dataset is not spatially complete and only provides data where HadEX2 had data? Or do you also use ERA-Interim to fill in spatially?

Page 5, S4, 4th to 3rd line from the bottom: "negative trends are lower" seems ambiguous: are they less strong / less negative, or less of an increase compared to another measure, for example?

Page 6, end of S4: you may want to acknowledge that there are some larger regional differences in particular the SUM for summer extremes does not display the distinct observed cooling trends over NW Europe and central Asia

Page 13 line 5: remove "." after "variability"

Reviewer #3 (Remarks to the Author):

This paper investigates the observed trends in DJFM cold and JJAS warm extremes over the 1950-2014 period, with a focus on the 2002-2014 period of the hiatus in global warming. While the rise in global temperature has slowed down, the occurrence of cold and warm extremes, in winter and summer respectively, has increased over land. The authors show that skillful predictors of the global temperature variability (ENSO/GHGs/aerosols/solar irradiance) fail to explain the interannual/interdecadal variability in temperature extremes. They seek other predictors that could explain the cold and warm extremes variability. They identify two distinct patterns for the two

seasons that improve the skill of the linear statistical model when included in the regression pool : in winter, a pattern of Z500 that resembles the negative NAO/NAM, and a SST pattern that resembles the AMO in summer. The observational evidences are completed by similar analyses of extreme temperature variability in a long coupled ocean-atmosphere control simulation. The increased trend in cold extremes is shown to be an extreme realization when compared to internal variability of the model. A potential link with Arctic sea ice loss is evoked that could explain its emergence in recent years in observations. In contrast, the summer warm extremes trend is in the range of internal climate variability. Sensitivity experiments performed with the CESM climate model support the role of the SST pattern in driving the increase in summer warm extremes in the recent decades.

The paper is clear and well-written, and presents some robust statistical analyses and well-designed numerical experiments. It provides a nice overview of the trends in temperature extremes over land in recent decades, and a convincing illustration of how temperature extremes can rise in a period of global warming slowdown, especially due to atmospheric internal variability.

I think this paper will be a very valuable contribution to the field and I'll fully support publication for it provided that the authors verify a couple of points as listed below. I require a major revision for the paper, but it is rather minor to me, it's just that I'd like to see the author's answers before I fully support publication.

Main comments

- I wonder about the robustness of the Z500/SST patterns you identify as precursors of the temperature extremes. They are defined from regressing the temperature indices on the gridded Z500/SST anomalies over 1950-2014, but is this strongly dependent to the time period that is chosen ? If you were using a training period to identify the patterns with the partial regression analysis (for example the first half of the record, 1950-1980), then apply the linear model to predict the temperature indices over the latter period (1981-2014), would the results remain robust ? I guess the full period is needed to identify the patterns since a large fraction of the skill comes from their decadal/multidecadal variability rather than interannual variability (negative-NAO trend in the 2000's, and AMO cycle for the SST). The robustness of the patterns in regard of the period that is used should be discussed somewhere in the paper, or in the methods section.
- l. 85, what are the spatial and temporal correlations between the Z500 pattern and the NAO/NAM ? If it's high, why not using a NAO/NAM index directly ? Please justify the benefit of using the Z500 pattern. Same remark with the SST pattern, how is it correlated with the summer AMO, and why not using directly an AMO index ?
- The cold extreme trend is at the tail of the cold days trend distribution from the FLOR simulation. You mention that this simulation exhibit substantial long-term variability despite constant radiative forcing, but is it comparable to observations ? It would be nice to see a comparison of the PDF of cold and warm extremes in FLOR vs observations to support this claim.
- In link with my previous comment, how large is the variability of the Z500 pattern in FLOR, compared to observations ? Since the pattern resembles the NAO, you could plot a power spectra of the NAO index in your model and in observations to verify whether the model exhibits enough long-term NAO variability compared to observations. Recent studies have shown that the low-frequency variability of the NAO is too weak in current GCMs, which can lead to underestimated internal variability, especially in the North Atlantic region (e.g., Wang et al. 2017). It is possible that the low-frequency fluctuations of the NAO is underestimated in the FLOR simulation, which could partly explain why cold extremes trends are less tied to the Z500 pattern in FLOR than in the real world (section S6). Similarly, it would be nice to see a comparison of the power spectra of the AMV (or summer SST pattern) as simulated by FLOR vs observations.

Wang et al. (2017) NAO and its relationship with the Northern Hemisphere mean surface temperature in CMIP5 simulations. Journal of Geophys. Res., DOI:10.1002/2016JD025979

Minor comments

- l. 45 : you refer to the trend as "time", which I don't find very clear. You could clarify here that your time predictor is the linear time trend (adding "referred to as time in the rest of the study", for instance). Or replace "time" by "trend" for the name of the predictor ?
- l. 233 : extremese -> extremes
- l. 497-501 and in other sections of the paper : when you refer to lags, you don't specify if they are negative or positive. Please clarify, maybe adding the sign of the lag (-11 months for instance)
- section S4, l. 10 : "at each grid **point** on predictor i"

Response to Reviewers

We thank Reviewer #1 for the constructive comments that have allowed us to clarify many points and to strengthen the revised manuscript. We respond to each comment below.

R1: *The current study does not add any new fundamental inside in the current understanding of the regional nature of temperature change over Northern Hemisphere land during the hiatus period.*

1a. Specifically, the role of the NAO during the hiatus period is described before (e.g. Trenberth et al. 2014, Nature Climate Change). The Trenberth et al. study highlights the role of the NAO during the cold winter of 2009-2010, 2010-2011 and 2012-2013 at the end of the hiatus period that are contributing to a negative cooling trend over Eurasia and Eastern United States. The Trenberth et al. study also suggests a mechanism of tropical Pacific origin that affects NAO/cold Eurasian temperatures and Arctic warming. Furthermore, the Trenberth et al. study also highlights the role of the PDO affecting specifically the cooling signal over western United States. Taking into account that the regression model outcome applied in the submitted study is very similar for seasonal means and extremes occurrence, it is not clear what fundamental new findings are provided.

Response: We acknowledge that in the original manuscript we did not highlight the new fundamental insights as clearly as we should have, but we believe that this study offers several novel findings, most notably:

- 1) Both wintertime TX10d and summertime TX90d increased in the Northern Hemisphere during the hiatus, a result that is counter-intuitive.**
- 2) The three important indices—annual-mean GMST, NH land winter SAT/TX10d and summer SAT/TX90d—are each governed by a different mechanism. The insight is enabled by examining these indices together for the first time in a single study.**

Although previous studies have touched on some of these elements in one way or another, we believe that we identify and synthesize several key unique elements that explain these two results, as we argue in our responses below.

We appreciate the reference to the Trenberth et al. (2014) study, and we agree that we should clarify the distinctions between our findings and theirs. We argue that our results are quite distinct from Trenberth in several key respects. Based on our investigation reported below, we find the following distinguishing results with respect to the role of the NAO during the hiatus period:

- Although the z500 extremes pattern that we identify is related to the NAO, it is not equivalent to the canonical NAO and is more closely connected to hemispheric land cold extreme occurrences than the NAO.**

- There is no evidence that tropical Pacific SSTs during the hiatus period excited the z500 extremes pattern, which contrasts the claim of Trenberth et al. (2014).

To address the first point above, which also addresses a comment raised by Reviewer 3, we first determine how closely the z500 extremes pattern is related to the NAO. We downloaded the monthly mean NAO index from the NOAA Climate Prediction Center website and found that the correlation coefficient between the DJFM z500 extremes index and NAO index is -0.65. Clearly the two indices are significantly related but there also is a substantial amount of z500 extremes index variability that cannot be explained by the NAO.

We next address how well the NAO index can be used as a substitute for the z500 extremes index to explain hemispheric land cold extreme and seasonal mean temperature variability. After linearly removing the influence of the other TX10d predictors (time trend, ENSO, and volcanic AOD), the z500 extremes index can explain 56% of the residual DJFM TX10d variance. The NAO index, in contrast, only explains 17% of the residual TX10d variance. *This finding indicates that the NAO index is not an adequate substitute for the z500 extremes index that we have identified, and that the z500 extremes index is much more closely related to NH land cold extreme occurrences than the NAO index.* This suggests that within the continuum of NAO-like atmospheric circulation patterns, there is considerable variation in the strength of the relationship with hemispheric extreme temperature occurrences and that, for the purposes of this study, it is important to identify the NAO-like pattern that is most strongly related to extreme temperature occurrences. In addition, this result indicates that the canonical NAO cannot explain the increase in hemispheric cold extreme occurrences during the hiatus period like the z500 extremes pattern can.

Similarly, we compared the relationship between the two indices and seasonal mean hemispheric land temperature. As reported in the original version of the manuscript, the correlation between the z500 extremes index and linearly detrended DJFM NH land temperature is -0.54. The correlation with the NAO index is only 0.19. Again, the link with the z500 extremes index clearly is much stronger. This finding relates to another comment by the reviewer about why we relate the z500 extremes pattern to the “Warm Arctic Cold Continents” pattern, which we discuss in a later reply.

Overall, these findings indicate that although Trenberth et al. (2014) and others may have suggested a connection between the NAO and recent increases in cold air outbreaks in some regions such as Europe, the connection with the NAO is not as simple as it may appear or may have been implied in previous studies. These previous studies have not performed as rigorous or as quantitative an analysis of the predictors of hemispheric cold extreme occurrences as we have performed here. Our study examines many factors that have been known to impact both global and regional climate variability, but we specifically

single out the patterns that dominate seasonal, hemispheric land temperature variability, which have notable contrasts to canonical patterns identified previously.

As mentioned by the reviewer, Trenberth et al. (2014) also suggest that the predominantly negative phase of the NAO and the outbreaks of cold over Europe were forced by the tropical Pacific through quasi-stationary atmospheric Rossby waves. Although we believe this is an interesting hypothesis, we also believe that this claim is debatable. The link between tropical Pacific SSTs and the NAO has been highly debated, but the consensus is that the negative phase of the NAO is associated with *warm* equatorial tropical Pacific conditions (e.g. Brönnimann 2007, doi:10.1029/2006RG000199). Such conditions strongly contrast the La Niña-like SSTs that characterized the hiatus period. Indeed, one of the motivators of our study and our approach was that the climate model simulations of Kosaka and Xie (2013) for which the equatorial tropical Pacific SSTs are restored to observed values could not explain the continental wintertime cooling during the hiatus, particularly over Eurasia.

To investigate the connection between the z500 extremes pattern and tropical SSTs, we calculated the partial regression coefficients of the z500 extremes index on the global SST anomalies. This calculation, illustrated below and in Fig. S10 of the revised manuscript, reveals that the z500 extremes pattern is weakly though significantly related to positive SST anomalies in the tropical equatorial Pacific.

Figure R1.1: Partial regressions of December – March SST anomalies ($^{\circ}\text{C}$) on the December – March z500 extremes index. Stippling indicates statistically significant regression coefficients at the 5% significance level.

This result is consistent with the body of literature suggesting a link between El Niño and the negative phase of the NAO. The SST pattern above, however, clearly contrasts the cool

equatorial Pacific conditions that characterized the hiatus period, suggesting that the prevalence of the z500 extremes pattern during the hiatus was not attributable to the pattern of tropical Pacific SSTs. Therefore, our findings contrast the claim of Trenberth et al. (2014) mentioned above. We do not disagree that the pattern of tropical Pacific SST anomalies during the hiatus period may have impacted some facets of the North Atlantic atmospheric circulation, as argued by Trenberth et al. (2014) and others (e.g. Ding et al. 2014, *Nature*), but we do not find any convincing evidence that tropical Pacific SSTs during the hiatus pattern could have forced the NAO-like z500 extremes pattern that was responsible for frequent cold outbreaks over Eurasia.

We believe that this investigation motivated by the reviewer's comment is an important addition to the revised manuscript. In lines 98-102, we now discuss the relationship between the z500 extremes pattern and the NAO and AO, while noting that the z500 extremes pattern is more closely tied to cold extreme occurrences. In Section S6 of the supplementary material, we discuss this analysis in detail. In lines 184-192, we discuss the relationship between the z500 extremes pattern and SSTs. We discuss the SST partial regression pattern in Section S7.

R1: *1b. The role of AMO in the hiatus has been discussed before (See Medhough et al. 2017 and references therein). The authors of the submitted study to not reconcile their findings with the findings in the literature on this topic. Furthermore, the described results of CESM1 experiments showing trends for the period 1979-2012 do not explain the increased in hot extremes during the period 2002-2014. How has the AMO changed during that period superposed on the anthropogenic signal?*

Response: We agree that we should improve the discussion of the identified SST extremes pattern in the context of the AMO and recent SST trends. First, we agree that other studies including Medhough et al. (2017) discuss the role of the AMO during the hiatus period, but as far as we are aware, those studies that have not focused on summertime land temperature or extreme temperature occurrence. Therefore, we believe that our study is distinct from those that focus more on the role of the AMO on the global mean temperature slowdown.

Next, we clarify the relationship between the SST extremes pattern and the AMO, just as we clarified the relationship between the z500 extremes pattern and the NAO. The correlation between the MJJ SST extremes index and the AMO index (Trenberth and Shea 2006) is 0.66. (We obtain similar results if we use the alternative AMO definition of van Oldenborgh et al. 2009). Therefore, similar to the arguments raised above, the SST extremes index is significantly related to the AMO but there is substantial variability of the SST extremes pattern that cannot be explained by the AMO. The Pacific component of the

pattern, which, as mentioned in the main text, resembles the “Pacific Extreme Pattern” (McKinnon et al. 2016), appears to be an important component of the SST extremes component that is unique from the AMO despite the prominence of Atlantic SST anomalies in the SST extremes pattern.

As in the analysis of the z500 extremes pattern and the NAO index, we find that the SST extremes index is more closely related to hemispheric land extreme temperature occurrence and seasonal mean temperature than the AMO. Following the same procedure described in our first response, we find that the SST extremes index can explain 75% of the residual variance of JJAS TX90d. The AMO only explains 35% of the residual variance. Therefore, we have identified a previously undocumented pattern that explains the variation in summertime extreme temperature occurrence better than the AMO index. In addition, we document in the original version of the manuscript that the correlation between the SST extremes index and the linearly detrended JJAS NH land temperature is 0.68. The correlation for the AMO is only 0.20. In summary, although the SST extremes index is significantly related to the AMO, it is much more closely tied to hemispheric extreme temperature occurrence and hemispheric, seasonal mean temperature.

The reviewer raises a good point that we should make a clearer connection between the 1979-2012 trend used for the CESM experiments and the SST extremes pattern during the 2002-2014 period. As shown in Figs. 2d and 3d of the manuscript, the SST extremes index featured a positive trend beginning ~1979 and continuing through the hiatus period. In addition, the annual mean SST trend from 1979-2012 bears a strong resemblance to both the MJJ SST anomaly from 2002-2014 (pattern correlation of 0.74) and the SST extremes pattern (pattern correlation of 0.55) (see below).

Figure R1.2: (top) Annual mean 1979-2012 SST trend ($^{\circ}\text{C} [34 \text{ yr}]^{-1}$) and (bottom) MJJ 2002-2014 SST anomaly ($^{\circ}\text{C}$) relative to the 1951-2000 base period.

Therefore, the annual mean 1979-2012 SST trend pattern used in the CESM1 experiments, the MJJ 2002-2014 SST anomaly, and the SST extremes pattern are closely related. Consequently, we argue that the CESM1 experiments indicate a more general connection between the AMO-like SST extremes pattern and multi-decadal trends in summertime land temperatures.

We believe that these clarifications in response to the reviewer’s comment have strengthened the revised manuscript. We clarify the relationship between the SST extremes pattern and AMO and the uniqueness of the SST extremes pattern in lines 119-124 of the main text and in Supplementary Section S6. We also discuss the similarity among the 1979-2012 SST trend pattern, the SST extremes pattern, and the 2002-2014 May – June SST anomaly pattern in lines 222-223 and 726-728 in the main text.

Overall, we believe that these responses to Reviewer 1’s first two comments demonstrate that our study does provide unique insights into the sources of regional temperature variability. We provide a focus on drivers of hemispheric land temperature variability that have contrasted the global mean temperature focus in most hiatus studies, we have demonstrated that we have identified unique patterns that are more closely tied to hemispheric extreme temperature occurrence than canonical indices, and we challenge some existing hypotheses about regional temperature variability during the recent hiatus period.

R1: 2. *Definition of the hiatus period: The authors of this submitted study define the hiatus period from 2002-2014. This period strongly differs from definitions in the literatures using the period 1998-2012 or 1998-2013 (e.g., Medhough et al. 2017, Nature; Trenberth et al. 2014, Nature Climate Change). The reasoning for using the period 2002-2014 is not given.*

Response: We chose the 2002-2014 period following Kosaka and Xie (2013), a highly cited study that defined the hiatus as the 2002-2012 period. We extended the period to 2014 as more data became available following their study, and that year seems like a natural end point since it precedes the extreme El Niño of 2015/16. The results of our study, however, are not sensitive to the particular definition of hiatus period. The time series illustrated in this study (Figs. 1-3) demonstrate that the variations over the 1998-2012 or 1998-2013 period would be qualitatively similar. For example, the ERA-Interim DJFM TX10d and JJAS TX90d linear trends are both positive over the 1998-2012 period (0.4 and 3.2 d [10yr]⁻¹, respectively).

In the revised manuscript, we now note in the main text that our conclusions are not sensitive to the choice of hiatus period (lines 45-46) and we give a brief rationale for choosing the 2002-2014 period in the Methods section (lines 568-573).

R1: 3. *Utilizing of the term “Warm-Arctic –Cold Continent” (WACC) circulation pattern: The authors do not provide any evidence for the occurrence and importance of such a prescriptive term for the circulation pattern instead of referring to the NAO which resembles their identified 500hPa extremes circulation pattern (as the authors point out themselves). The NAO which is a well-defined mode of variability varies strongly on decadal to multi-decadal time scale with well-established effects on the climate on Northern Hemisphere continental climate. Furthermore, it becomes obvious that recent negative NAO events that occurred during the second half of the studied period contributed to their result. Finally, the physical mechanism of such a WACC pattern that is different from the NAO itself is not clear. Other recent studies point to the impact of a different circulation phenomenon, namely the strengthening of the Siberian high as a main driver of the Eurasian cooling trend over the 1990 to 2014 period when the so called WACC like temperature trend pattern occurred (Sun et al. 2016, GRL).*

Response: We recognize this comment as a valid criticism of the original version of the manuscript, and we hope that our response to the reviewer’s first comment clarifies why distinguish the z500 extremes pattern from the NAO and also make a connection to a “Warm-Arctic-Cold-Continents” pattern. In summary, (1) although the z500 extremes index is significantly related to the NAO index, the NAO index linearly explains less than half of the z500 extremes index; (2) the z500 extremes index explains substantially more variance in the hemispheric cold extreme occurrences than the NAO index; and (3) most

specifically in relation to the connection with the “WACC” pattern, the z500 extremes index is much more highly correlated with seasonal mean hemispheric continental temperature than the NAO index. These points are now explained more clearly in the main text and Supplementary Section S6.

Regarding the physical mechanisms that distinguish the pattern we identify and the NAO, we recognize that we have more questions than answers at this point. However, we believe that our study provides a starting point for future studies to address these questions. There is recent work that recognizes the Arctic Oscillation as a continuum of patterns with unique dynamical mechanisms rather than a single mode (Dai and Tan 2017, doi: 10.1175/JCLI-D-16-0467.1). It seems plausible that the WACC-related z500 extremes pattern is part of this NAO/AO continuum that happens to be more strongly related to hemispheric land temperature than other members of the continuum.

RI: As a whole, investigating trends in temperature extreme occurrences averaged of the Northern Hemisphere land instead of seasonal mean temperature trends does not provide any additional insight into regional land temperature changes during the hiatus period specially over such a very short record. Specifically, these short time series (13 years) of extreme occurrence are very noisy and regression lines to illustrate trends are strongly affected by single years.

Response: Regarding the reviewer’s first point, we agree that that there is a clear connection between extreme temperature occurrences and seasonal mean temperature trends over land. Following comments by Reviewer 2, we have strengthened this discussion in the manuscript, particularly in supplementary section S5. However, we argue that our manuscript provides unique insights into the climate patterns that dominate the variability of both extreme temperature occurrence and seasonal mean temperature over Northern Hemisphere land, which are distinct from those that dominate annual mean, global mean surface temperature. As we argue in our responses above, the patterns we identify and the arguments we make about the sources of regional temperature variability are distinct from the patterns and arguments presented in previous studies.

We also agree with the reviewer’s second point that short time series are noisy, and trends defined over such short periods are not robust. We attempt to bring out this point in our analysis related to Fig. 4. However, we believe that there is both a scientific and societal need to be able to explain these approximately decadal climate variations, even if they may be considered noise from an anthropogenically forced climate change perspective. Indeed, this spirit appears to have been an important driving force in the many global warming hiatus studies over the past few years. This idea is echoed in the Trenberth et al. (2014) study referenced by the reviewer, as they write that “it is vital to understand related interannual and decadal variability... and its regionality.” We believe our study

contributes to this objective by providing insights on regional temperature variations that are much more societally relevant than global mean temperature.

We thank Reviewer #2 for the thorough and constructive comments that have resulted in a strengthened manuscript. We respond to each comment below.

R2: Also the annual global land average temperatures continued to increase during the ‘hiatus’ period (see e.g. Seneviratne et al 2014, doi:10.1038/nclimate2145). So how do these increases in summer hot extreme temperatures differ from the increasing average temperatures over land?

Response: The increases in warm and cold extreme occurrences do indeed closely track the area-averaged seasonal mean land temperature. The following figure shows the time series of the seasonal land temperature anomaly, now shown as Supplementary Fig. S7, which can be compared with Fig. 1a. The correlation coefficients between DJFM TX10d and seasonal mean temperature time series is -0.86. The correspondence is even stronger in boreal summer, as the correlation for JJAS TX90d is 0.96. Following the reviewer’s comment, we now mention this point in lines 240-243 of the main text.

Figure R2.1: Time series of area-averaged December – March (blue) and June – September (red) NH land surface air temperature anomalies (K) since 1979.

As indicated by the reviewer’s next comment, the correspondence may be particularly strong because of the choice of a relatively moderate threshold for extremes. However, these results are in line with cited works (Rhines and Huybers 2013; Argüeso et al. 2016) that note the close connection between changes in mean and extreme temperatures. These results also fit within the overall theme of our study that the large-scale climate patterns responsible for variations in both mean and extreme temperature occurrence over

Northern Hemisphere continents are distinct from those of global mean surface temperature (lines 235-255).

R2: *Cohen et al. 2012 (doi: 10.1029/2011GL050582) documented seasonal asymmetries in temperature change (incl. summer vs winter). The authors included a brief discussion around differences between seasonal means and seasonal extremes, but I think it could be clarified in how far these ‘extreme’ changes relate or possibly exceed the mean changes. Given the authors use measures of relatively moderate extremes (that occur on average on 10% of the days), I expect the results between these extremes and the seasonal averages to be reasonably similar. However, there may be larger differences when looking at some more extreme measures of hot/cold extremes.*

Response: As the reviewer suggests and as the previous response indicates, the correspondence between changes in extreme temperature occurrences and seasonal mean temperature is strong, particularly for the 10%/90% thresholds that we have chosen. We thank the reviewer for mentioning the Cohen et al. (2012) study, as we agree that this study should be noted in the manuscript. In the revised version, we have restructured the paragraph from lines 235-245 to clarify how our work extends beyond the work of Cohen et al. (2012). That study suggested that the source of continental wintertime cooling also is the driver in the recent global warming slowdown; however, subsequent studies have demonstrated the importance of equatorial Pacific SSTs for the global warming slowdown, which do not have a strong connection with continental cooling, especially over Eurasia. Our work demonstrates the sources of these asymmetric trends in both temperature extreme occurrences and seasonal mean temperature, which contrast the dominant drivers of global mean surface temperature.

Regarding the possibility that more ‘extreme extremes’ may behave differently, we examined the time series of different measures of extreme temperatures following the methodology of Seneviratne et al. (2014). The key differences with that study are: (1) we consider both hot and cold extremes, (2) we partition the year into boreal summer (June – September) and boreal winter (December – March), and (3) we focus only on the Northern Hemisphere. These results (Fig. R2.2 below and Fig. S15 in the revised manuscript) indicate, as in Seneviratne et al. (2014) for annual global data, that the hottest Northern Hemisphere summer extremes have increased faster than the more moderate hot extremes. For cold extremes, however, we do not see such an amplification of the trend in the coldest extremes.

Figure R2.2: Time series of the ratio of NH land area affected by the exceedances of 5, 10, and 20 (a) December – March extreme cold days and (b) June – September extreme warm days relative to the 1979-2010 average. The dashed black line at a land area ratio of 1 corresponds with the 1979-2010 average. The dashed colored lines indicate the 2002-2014 linear trend lines; the 2002-2014 trends for cold (warm) extreme exceedances are 0.037 (0.031), 0.055 (0.060), and 0.061 (0.144) $[10 \text{ yrs}]^{-1}$ for ExD5, ExD10, and ExD20, respectively.

We now discuss these findings in Supplementary Section S9 in the manuscript. We address the reviewer’s main point in the last paragraph: “These findings suggest that the strong correspondence between seasonal mean temperature and extreme temperature occurrences during the hiatus period found in this study may not hold as well for more extreme measures of summertime hot extremes but is likely to be robust for wintertime cold extremes. The reason for the amplified response of the hottest extremes requires more study, but soil moisture-temperature feedbacks are one plausible culprit (Vogel et al. 2017), although there is some indication that this mechanism for accelerated warming of hot extremes in climate models may not hold in observations, at least in some regions (Donat et al. 2017, doi:10.1002/2017GL073733). Overall, the degree to which extreme temperatures follow the changes in the mean of the temperature distribution may depend on the region, season, type of extreme (hot or cold), and the threshold used to define the extremes.”

R2: As a conclusion, the authors may want to highlight that GMST probably is not a particularly useful measure for relevant climate states or of climate change in general, as it may average out different kinds of events, seasonal and regional characteristics.

Response: We agree with this perspective and thank the reviewer for the suggestion. We now bring out the suggested point in lines 246-249 of the revised manuscript.

R2: Line 7: it should be specified if the increase is e.g. in frequency or intensity or associated temperature. In particular for cold extremes the term “increase” can be ambiguous: increasing frequency would be consistent with cooler conditions, increase in associated temperatures with warming.

Response: We agree that our original phrasing was ambiguous and how now changed the word “extremes” to “extreme occurrences” in line 8.

R2: Line 93/94: do you really mean “anomalous warming”, or rather “anomalously warm” (i.e. displaying a warm anomaly but not necessarily intensifying)?

Response: We agree that “warmth” rather than “warming” is a more accurate term, and so we have changed the text accordingly.

R2: Line 116: please specify how you “examine the significance”

Response: We now specify that we are referring to statistical significance, and that this is done through an analysis of the trend confidence intervals.

R2: Line 117: specify that “high resolution” refers to the atmospheric model, otherwise it seems in contradiction to “Low Resolution” in the following line

Response: We agree with the reviewer and have modified the text accordingly.

R2: Line 125: not clear what exactly “this conclusion” refers to

Response: We agree – “this conclusion” has been changed to “the unusual nature of these recent trends.”

R2: Line 144: it would be preferable to reserve the use of “significant” when you mean “statistically significant” (and then also explicitly specify the statistic meaning in those occurrences).

Response: We agree with this perspective, and in this case, we intended a statistical meaning, although we did not explain clearly. We determine that the JJAS TX90d trend has accelerated because the 95% confidence intervals of the linear trends with start years after ~1965 do not contain the linear trend for the full period (until later in the record when the sample sizes for the trend calculations become much smaller). In the revised manuscript we have clarified our meaning by modifying the sentence as: “This finding indicates a significant acceleration of the positive trend in warm extreme occurrences from the mid to late 20th century, as determined by the failure of the 95% confidence intervals of the linear trends with start years after ~1965 to contain the 1951-2014 linear trend.”

R2: Line 182: better “levelled” than “level”?

Response: We agree about improving the word choice and have changed “level” to “little change in.”

R2: Line 186: “occurrence of...occurrences” – better reword?

Response: We have shortened the sentence and removed the redundancy of “occurrence.”

R2: Line 190/191: *The causality is not clear: it seems like you are saying the (relatively small) size of NH land area is responsible for its larger variability? It should also be clarified “larger” than what?*

Response: We agree that this sentence was confusing and have removed it during the revisions described in earlier responses.

R2: Line 417: *was the p-value / degrees of freedom corrected for auto-correlation in the fields (see Wilks 2016, <https://doi.org/10.1175/BAMS-D-15-00267.1>)?*

Response: Yes, the degrees of freedom have been adjusted for autocorrelation. We specify the method in lines 669-671 of the manuscript. Following the reviewer’s comment, we have revised the caption to state “stippling indicates regression coefficients that are statistically significant at the 5% level based on a two-sided *t*-test for which the temporal degrees of freedom are adjusted for autocorrelation (see Methods).”

R2: Line 436: *“surface temperature” – specify it is the average temperature*

Response: Fixed, thank you.

R2: Line 448: *The “satellite era” continues until present, so it would be good to specify why the analysis ends in 2014. Presumably because GMST increased after this?*

Response: Following another comment by Reviewer 1, we now have clarified the reasons why we define the hiatus period the way we have. We followed the definition used in Kosaka and Xie (2013) but then extended the data to 2014 as more data became available. As the reviewer suggests, we ended in 2014 because of the strong El Niño and associated rise in global mean temperatures that followed. We specify our reasoning for the period we have chosen in lines 568-573 of the revised methods section.

R2: *Comments on the Supplementary information*

Title: the title is different to the title of the main text

Response: Thank you for pointing out this oversight – the title has been fixed.

R2: *S2: Please specify if also the merged dataset is not spatially complete and only provides data where HadEX2 had data? Or do you also use ERA-Interim to fill in spatially?*

Response: We only include regions for which HadEX2 has at least 90% temporal coverage, and so we do not use ERA-Interim for spatial interpolation. We now specify this point in

the text. We also indicate in the caption of Fig. S2 that darker gray indicates land regions not included in the merged dataset.

R2: *Page 5, S4, 4th to 3rd line from the bottom: “negative trends are lower” seems ambiguous: are they less strong / less negative, or less of an increase compared to another measure, for example?*

Response: **We agree that the sentence was poorly constructed. We have revised accordingly.**

R2: *Page 6, end of S4: you may want to acknowledge that there are some larger regional differences in particular the SUM for summer extremes does not display the distinct observed cooling trends over NW Europe and central Asia*

Response: **This is a fair point and one worth mentioning. We have revised the text accordingly.**

R2: *Page 13 line 5: remove “.” after “variability”*

Response: **Fixed, thank you.**

We thank Reviewer #3 for the insightful and helpful comments that have resulted in a strengthened manuscript. We respond to each comment below.

R3: - *I wonder about the robustness of the Z500/SST patterns you identify as precursors of the temperature extremes. They are defined from regressing the temperature indices on the gridded Z500/SST anomalies over 1950-2014, but is this strongly dependent to the time period that is chosen ? If you were using a training period to identify the patterns with the partial regression analysis (for example the first half of the record, 1950-1980), then apply the linear model to predict the temperature indices over the latter period (1981-2014), would the results remain robust ? I guess the full period is needed to identify the patterns since a large fraction of the skill comes from their decadal/multidecadal variability rather than interannual variability (negative-NAO trend in the 2000's, and AMO cycle for the SST). The robustness of the patterns in regard of the period that is used should be discussed somewhere in the paper, or in the methods section.*

Response: We agree that the robustness of the z500 and SST extremes patterns deserves additional consideration. As the reviewer indicates, it may be too strict a test to divide the record in half, given that the sample sizes in the training period would be quite small, and there is substantial multidecadal variability that we would miss in the training period. However, because we are focusing on the 2002-2014 period, we believe it is fair to examine if the patterns and indices are robust if we exclude the 2002-2014 period from the training set.

For the tests we performed, we split the data into the 1951-2001 training period and a 2002-2014 validation period. We then removed the influence of the time trend, ENSO, and volcanic AOD from the 2002-2014 DJFM TX10d, JJAS TX90d, the gridded SST, and the gridded z500 fields, just as in the original analysis, but here all regression coefficients were determined from the 1951-2001 training period. We then performed PLSR analysis to determine the z500 extremes pattern, SST extremes pattern, and regression coefficients of the residual TX10d and TX90d onto the corresponding extremes pattern index with the 1951-2001 data. Finally, we predicted the z500 and SST extremes pattern contributions to TX10d and TX90d, respectively, during the 2002-2014 period. We then compared these out-of-sample calculations with the in-sample calculations reported in the manuscript. In summary, there are three potential sources of disagreement between the in-sample and out-of-sample calculations: (1) differences in the ENSO, time trend, and AOD regression coefficients that affect the adjustment of the TX10d and TX90d time series prior to the PLSR analysis; (2) differences in the z500 and SST extremes patterns between the two different datasets, and (3) differences in the z500 and SST extremes pattern regression coefficients, holding the z500 and SST patterns identical for the two datasets.

The top two panels in the figure below show the z500 and SST extremes patterns determined from the 1951-2001 data. As we can see, the patterns are very similar to those reported in the manuscript. This analysis confirms that the patterns and their relationships with extreme temperature occurrence were not unique to the hiatus period. The bottom two panels show the out-of-sample predictions of the z500 and SST extremes pattern contributions to DJFM TX10d and JJAS TX90d, respectively, in comparison with the in-sample partial regressions reported in the manuscript (Fig. 2). We see some differences between the in-sample and out-of-sample time series, but the hiatus period trends and much of the interannual variability are in good agreement. Overall, we believe that this analysis supports the robustness of the z500 and SST extremes patterns and their contribution to Northern Hemisphere extreme temperature variability during the hiatus period.

Figure R3.1: The (a) z500 and (b) SST extremes patterns, shown as partial regression maps (units of m and °C, respectively) and calculated in the same way as in Fig. 3 but from 1951-2001 data only. The in-sample partial regressions of (c) DJFM TX10d on the z500 extremes index and (d) JJAS TX90d on the SST extremes index for the 2002-2014 hiatus period are shown as solid blue lines (same values as in Fig. 2). The predicted contributions of the (c) z500 and (d) SST extremes patterns to DJFM TX10d and JJAS TX90d, respectively, during the 2002-2014 hiatus period based on regression parameters determined from 1951-2001 data are shown as dashed blue lines.

We now explain the results of these calculation in the Methods section (lines 640-662). We also present the figure shown above as Supplementary Fig. S4.

R3: l. 85, what are the spatial and temporal correlations between the Z500 pattern and the NAO/NAM ? If it's high, why not using a NAO/NAM index directly ? Please justify the benefit of

using the Z500 pattern. Same remark with the SST pattern, how is it correlated with the summer AMO, and why not using directly an AMO index ?

Response: The reviewer's comment echoes a similar sentiment raised by Reviewer 1. In response, we have performed thorough comparisons between the z500 extremes index and NAO/AO indices and between the SST extremes index and AMO index. These results are now detailed in Supplementary Section S6. The key findings are:

- The z500 extremes index has a strong relationship with the NAO and AO index ($r = -0.65$ and -0.68 , respectively). The SST extremes index also has a strong relationship with the AMO index ($r = 0.66$). The significant relationships revealed above are not surprising, but they also reveal that there is a substantial amount of variability of the z500 and SST extremes patterns that cannot be explained by the NAO/AO or AMO.
- The z500 extremes pattern is much more strongly related to hemispheric cold extreme occurrences than either the NAO or AO. After linearly removing the influence of the other predictors (time trend, ENSO, and volcanic AOD) from the NH DJFM TX10d time series, the z500 extremes index explains 56% of the residual TX10d variance. The NAO and AO indices, however, only explain 17% of the residual TX10d variance.
- Similarly, the SST extremes pattern is much more strongly related to hemispheric warm extreme occurrences than the AMO. The SST extremes pattern explains 75% of the residual JJAS TX90d after the removal of the linear influence of the other predictors. The AMO index explains only 35% of the residual TX90d variance.

These findings demonstrate that, despite the similarity between the z500 and SST extremes patterns and canonical patterns of climate variability, there are key differences that connect the z500 and SST extremes patterns to extreme temperature occurrence more strongly.

R3: *The cold extreme trend is at the tail of the cold days trend distribution from the FLOR simulation. You mention that this simulation exhibit substantial long-term variability despite constant radiative forcing, but is it comparable to observations ? It would be nice to see a comparison of the PDF of cold and warm extremes in FLOR vs observations to support this claim.*

Response: This is a fair point. To address this comment and to keep the analysis consistent with that of the FLOR simulation, we calculated histograms of 13-yr TX10d and TX90d trends from the linearly detrended observational datasets. This analysis, illustrated below and now in Fig. S13 of the revised manuscript, suggests that the observed distribution of the observed TX10d and TX90d trends are wider than those of the FLOR simulation. When evaluating whether these distribution are significantly different from those of FLOR,

however, only the DJFM TX10d distributions are distinct at the 5% level on the basis of a two-sample Kolmogorov-Smirnov test.

Figure R3.2: Histogram of the (a) 13-yr DJFM TX10d and (b) 13-yr JJAS TX90d trends in the linearly detrended 1951-2014 observed datasets.

We now discuss these results in Supplementary Section S8 and lines 144-157 of the main text. We expand upon this result in response to the reviewer's next comment.

R3: *In link with my previous comment, how large is the variability of the Z500 pattern in FLOR, compared to observations? Since the pattern resembles the NAO, you could plot a power spectra of the NAO index in your model and in observations to verify whether the model exhibits enough long-term NAO variability compared to observations. Recent studies have shown that the low-frequency variability of the NAO is too weak in current GCMs, which can lead to underestimated internal variability, especially in the North Atlantic region (e.g., Wang et al. 2017). It is possible that the low-frequency fluctuations of the NAO is underestimated in the FLOR simulation, which could partly explain why cold extremes trends are less tied to the Z500 pattern in FLOR than in the real world (section S6). Similarly, it would be nice to see a comparison of the power spectra of the AMV (or summer SST pattern) as simulated by FLOR vs observations.*

Wang et al. (2017) NAO and its relationship with the Northern Hemisphere mean surface temperature in CMIP5 simulations. *Journal of Geophys. Res.*, DOI:10.1002/2016JD025979

Response: In light of the results presented above, we agree that it is worthwhile to investigate low-frequency variability in FLOR more closely. We followed the reviewer’s suggestion and calculated power spectra of the NAO in observations (reanalysis) and in FLOR. These calculations indicated that FLOR may underestimate low-frequency NAO variability, as suggested by the reviewer. However, given that there are some differences between the NAO and z500 extremes pattern, and between the observed and simulated NAO, we decided that it would be simpler to show maps of 500 hPa geopotential height standard deviations in the FLOR simulation and in linearly detrended reanalysis data. We show below and in Fig. S14 of the revised manuscript the standard deviations for both seasonal and 13-yr running mean data.

Figure R3.3: Standard deviations (m) of linearly detrended December – March 500 hPa geopotential height in (a) NCEP/NCAR reanalysis data and (b) the 500-yr FLOR simulation. (c,d) As in (a) and (b) but for 13-yr running mean 500 hPa geopotential height data. (e) Difference between reanalysis and FLOR standard deviations (a minus c). (f) Same as (e) but for 13-yr running mean data (b minus d).

Several features stand out in the plots shown above. First, on interannual timescales, FLOR overestimates the geopotential height variance in the northeast Pacific and southern North America (panel c). This feature is not unexpected, given that these regions are preferentially impacted by strong ENSO episodes, and that FLOR is known to overestimate ENSO variance (Vecchi et al. 2014). The second and more relevant feature is that FLOR underestimates geopotential height variance over the North Atlantic and parts of Eurasia, which is consistent with the reviewer’s suggestion. This feature particularly stands out in the low-frequency differences (panel f). Most interestingly, the two primary regions for which FLOR underestimates 500 hPa geopotential height variance correspond well with two action centers of the z500 extremes pattern (Fig. 3). As the reviewer suggests, this analysis indicates that climate models, including high-resolution models like FLOR, may underestimate natural, multidecadal variability of cold extreme occurrences owing to the underestimation of NAO-like variability over the North Atlantic and Eurasia.

We now discuss these findings in lines 158-171 of the main text and in Supplementary Section S8. We believe that this is a valuable addition to the manuscript because it demonstrates that previous studies that rely on current state-of-the-art climate models to attribute WACC-related changes may have a key deficiency in simulating multidecadal variability of continental cold extremes and related atmospheric circulation variability.

We have chosen not to follow a similar line of analysis with AMO-like variability and summertime warm extremes for several reasons. First, as described above, the distributions of 13-yr TX90d trends are not significantly different between FLOR and observations. Second, even if there are differences between FLOR and observations, it would be difficult to attribute its cause because, as discussed in the main text, we expect anthropogenic forcing, particularly from anthropogenic aerosols, to have some projection on the AMO and SST extremes pattern. Finally, even if FLOR underestimates internal Atlantic multidecadal variability, the implication would only strengthen the existing conclusion that internally driven, apparent accelerations of warm extreme occurrences like what occurred from 2002-2014 are relatively common in the climate model. For these reasons, we believe that additional analysis on AMO or AMO-like variability would contribute to the growing length of the manuscript without adding much additional clarity.

R3: *Minor comments*

- l. 45 : you refer to the trend as "time", which I don't find very clear. You could clarify here that your time predictor is the linear time trend (adding "referred to as time in the rest of the study", for instance). Or replace "time" by "trend" for the name of the predictor ?

Response: In the revised manuscript, we have replaced “time” with “time trend” following the reviewer’s suggestion.

R3: - l. 233 : *extremese -> extremes*

Response: Fixed, thank you.

R3: - l. 497-501 and in other sections of the paper : *when you refer to lags, you don't specify if they are negative or positive. Please clarify, maybe adding the sign of the lag (-11 months for instance)*

Response: We now specify all lags as negative when they indicate the predictor leading the predictand, and we clarify this convention in lines 591-592 of the methods section.

R3: - section S4, l. 10 : *"at each grid point on predictor i"*

Response: Fixed, thank you.

Reviewers' Comments:

Reviewer #1:

Remarks to the Author:

My concerns were properly addressed. I recommend the publication of this manuscript.

Reviewer #2:

Remarks to the Author:

The authors have satisfactorily addressed the reviewer comments raised. I still have a few additional suggestions for clarifications that I hope will help to further strengthen the manuscript.

Most importantly, the authors highlight at several places in the manuscript that the large-scale drivers related to variability of extreme temperatures are distinct from those related to GMST. I think this is confusing because the GMST, by definition, is affected by regional and seasonal temperature variability (including temperature extremes), as it averages over temperatures in space and time. Therefore the drivers identified here are also relevant to understanding GMST variability, and these aspects would be neglected in studies only focusing on GMST.

In turn, one could ask if the "distinct" factors used to explain the GMST 'hiatus' would also lead to a widening of the temperature distribution – i.e. would simulate a mean temperature 'hiatus' for the right reason and consistent with the observed one, which is characterized by increasing warm tail and decreasing cold tail of the temperature distribution compensating each other so that there was little change in the mean.

Specific comments

Line 36-38: The authors could add here that the GMST is just the average over regional and seasonal temperature variations. So understanding these will also better explain the GMST variability.

Line 43-45: in line with my general comment, it seems confusing that drivers of GMST and warm/cold tail variability would be "distinct" given GMST is by definition affected by warm and cold tail variability.

Line 52: please specify: which "century"

Line 72/73: Please insert "the frequency of" before "summertime warm extremes" and "NH land cold extremes"

Line 113-115 / Figure 2: I would appreciate if the authors could clarify some questions around the "time" time series: I am puzzled about the inter-annual variability in this predictor. For example, it is not clear to me why the predictor "time" (I assume independent of e.g. ENSO variability) would lead to more cold and fewer warm extremes in ~1998 compared to the previous 5 years. I.e. why is the "time" contribution not monotonous, what explains the year-to-year variability?

Also from a physical perspective, I wonder if e.g. GHG concentrations would be a more suitable predictor here, assuming the long-term warming during 1950-2014 (as shown here by increasing warm extremes and decreasing frequency of cold extremes) was primarily driven by increasing GHG as opposed to time?

Line 222: change "providing evidence" to "suggesting"

Line 275/276: Again, GMST is by definition related to regional and seasonal temperature variability, so it would not seem logical that the climate patterns are entirely distinct.

Line 297: Ref. 45 studies regional hot extremes relative to global mean temperatures. However, in the context of the discussion here, how different parts of the temperature PDF are changing, the authors may want to add reference to a study showing how hot extremes warm faster than the local mean temperatures (e.g. ref. 86: Donat et al 2017, doi:10.1002/2017GL073733) – i.e. consistent with a widening of the warm tail of the temperature PDF.

Line 385: check reference 27, repeats the title?

Reviewer #3:

None

Response to Reviewer #2

We thank Reviewer #2 for the careful reading of manuscript and the additional helpful comments. We respond to each comment below. All line numbers refer to the version of the manuscript with tracked changes (All Markup).

R2: Most importantly, the authors highlight at several places in the manuscript that the large-scale drivers related to variability of extreme temperatures are distinct from those related to GMST. I think this is confusing because the GMST, by definition, is affected by regional and seasonal temperature variability (including temperature extremes), as it averages over temperatures in space and time. Therefore the drivers identified here are also relevant to understanding GMST variability, and these aspects would be neglected in studies only focusing on GMST.

In turn, one could ask if the “distinct” factors used to explain the GMST ‘hiatus’ would also lead to a widening of the temperature distribution – i.e. would simulate a mean temperature ‘hiatus’ for the right reason and consistent with the observed one, which is characterized by increasing warm tail and decreasing cold tail of the temperature distribution compensating each other so that there was little change in the mean.

Response: We appreciate the reviewer pointing out this potential source of confusion. We agree that variations in GMST are impacted by the regional and seasonal temperature variations we discuss in this study. An example that is evident in our study is ENSO, which is a significant predictor of both GMST and extreme temperature occurrences over land. The primary point we attempt to make is that the *dominant* sources of annual GMST and continental extreme temperature variability are distinct, which implies the existence of mechanisms that preferentially warm or cool the land relative to the oceans or that modulate the seasonal evolution of global temperature. The reviewer’s comments have helped us to strengthen our Introduction, and we believe that these points are now made clearer in the third paragraph (lines 44-56). We also clarify this point in line 293 by changing “distinct from those of GMST” to “distinct from those that dominate the variability of annual GMST.”

R2: Specific comments

Line 36-38: The authors could add here that the GMST is just the average over regional and seasonal temperature variations. So understanding these will also better explain the GMST variability.

Response: We have chosen not to modify this section because the regional and seasonal NH land temperature variations will not *necessarily* help us to understand GMST variability,

given that Northern Hemisphere land represents a relatively small fraction of the global surface area.

R2: *Line 43-45: in line with my general comment, it seems confusing that drivers of GMST and warm/cold tail variability would be “distinct” given GMST is by definition affected by warm and cold tail variability.*

Response: We understand this point, but in this study we focus on warm and cold tail variability over Northern Hemisphere continents, which represent a relatively small fraction of global surface area but a large fraction of the impacts of extreme temperature occurrence. Therefore, it is conceivable that the dominant drivers of GMST and more impactful extreme temperatures are distinct if the warming/cooling of the NH continental regions are compensated by temperature changes of reduced magnitude or even opposite sign over the oceans and/or Southern Hemisphere. We hope that this point is now clearer in lines 44-56 of the Introduction.

R2: *Line 52: please specify: which “century”*

Response: Added, thank you!

R2: *Line 72/73: Please insert “the frequency of” before “summertime warm extremes” and “NH land cold extremes”*

Response: Added, thank you!

R2: *Line 113-115 / Figure 2: I would appreciate if the authors could clarify some questions around the “time” time series: I am puzzled about the inter-annual variability in this predictor. For example, it is not clear to me why the predictor “time” (I assume independent of e.g. ENSO variability) would lead to more cold and fewer warm extremes in ~1998 compared to the previous 5 years. I.e. why is the “time” contribution not monotonous, what explains the year-to-year variability?*

Response: The interannual variations with the time trend component relate to its small dependence on the other predictors. The predictors are not strongly correlated with each other, but the correlations are not zero (except for those of the z500 and SST extremes patterns by construction). The strongest predictor correlations occur between ENSO and time (~0.40) for DJFM TX10d. The time series in the bottom of Fig. 2 are constructed by subtracting the regressions with the predictor removed from the full regressions. In the case of the trend time series in Fig. 2, when time is removed as a predictor, the influence of the trend is carried in part by ENSO because of the non-zero correlation between the two. This leads to some of the interannual variations attributed to the trend. The motivation of

this calculation is to illustrate the influence of each predictor that is completely independent of all other predictors.

However, after considering the reviewer's comment, we realize that our original calculations are unnecessarily confusing. Given that collinearity is not strong, it is more straightforward to illustrate each component as the product of the predictor value and its corresponding regression coefficient. (We note that the z500 and SST extremes pattern components have not changed at all because these patterns are uncorrelated with all other predictors by construction.) We now use this decomposition in the revised Fig. 2.

Figure 2. Linear regressions of wintertime cold and summertime warm extremes. Time series of (a) wintertime cold and (b) summertime warm extreme temperature occurrence (d season⁻¹) over Northern Hemisphere land from 1951-2014 in observations (dark blue and red) and for linear regression models with four predictors (light blue and orange). Shading indicates the 95% confidence interval for the regression. (c,d) Contribution of each individual predictor for the frequency of (c) cold extreme and (d) warm extremes, calculated by multiplying the predictors by their corresponding regression coefficients. Red lines indicate linear trend lines for the hiatus period (2002-2014). Major volcanic eruptions are labeled in the AOD partial regression plots. The top regressions in (c) and (d) indicate the influence of the z500 and SST extremes patterns, respectively.

The main features have not changed from the original Fig. 2, and the time series are easier to interpret. The time trend components are monotonically decreasing and increasing, as

the reviewer expected, and now the sum of the components is equal to the full regression with the time mean removed. We thank the reviewer for prompting us to clarify this analysis.

The similarity between the previous and revised Fig. 2 is an indication that the predictors are not strongly collinear, which supports the interpretability of the analysis. We now include discussion of this point in lines 672-675 of the Methods section.

Also from a physical perspective, I wonder if e.g. GHG concentrations would be a more suitable predictor here, assuming the long-term warming during 1950-2014 (as shown here by increasing warm extremes and decreasing frequency of cold extremes) was primarily driven by increasing GHG as opposed to time?

Response: We considered this alternative but opted to keep time as a predictor for a couple of reasons. First, we could not include GHG concentrations without also including anthropogenic aerosol forcing, the latter of which has substantial uncertainty over the period of interest. Second, the regression models as currently constructed perform exceptionally well (Fig. 2), which indicates that the z500 and especially SST extremes patterns can capture any nonlinearity in the TX10d and TX90d trends. Additional synthesis allows us to further diagnose how both internal variability and radiative forcing may contribute to their variations and therefore to the nonlinearity in the TX10d and TX90d trends. Therefore, the current analysis allows to highlight how both radiative forcing and internal variability may have similar spatial fingerprints with respect to dominant predictors of Northern Hemisphere temperature extremes.

Because other readers may have the same concern, we have added discussion of this point in lines 649-660 of the Methods section in the revised manuscript.

R2: Line 222: change “providing evidence” to “suggesting”

Response: Done

R2: Line 275/276: Again, GMST is by definition related to regional and seasonal temperature variability, so it would not seem logical that the climate patterns are entirely distinct.

Response: As discussed above, we have modified this sentence by clarifying that the patterns that *dominate* the variability are distinguishable, although we agree that some factors (e.g., ENSO) can be important for both GMST and regionally varying seasonal temperature.

R2: *Line 297: Ref. 45 studies regional hot extremes relative to global mean temperatures. However, in the context of the discussion here, how different parts of the temperature PDF are changing, the authors may want to add reference to a study showing how hot extremes warm faster than the local mean temperatures (e.g. ref. 86: Donat et al 2017, doi:10.1002/2017GL073733) – i.e. consistent with a widening of the warm tail of the temperature PDF.*

Response: We agree that Donat et al. (2017) is an appropriate reference here, given their use of the local mean temperatures as a reference. We have added this reference to this section.

R2: *Line 385: check reference 27, repeats the title?*

Response: Fixed, thank you!